# FEDERATED COORDINATION: DISTRIBUTED AND PRIVATE STRATEGY ALIGNMENT

## ABSTRACT

Coordination in multi-agent systems is critical for optimizing collective outcomes and is applicable in diverse fields such as drone swarms, emergency response, and more. Despite extensive research, the distributed coordination strategy alignment problem—where all agents follow the same strategy and execute the prescribed actions without a global coordinator—remains largely unexplored, posing challenges in scalability and privacy preservation. We introduce a new research problem termed "federated coordination", which seeks to achieve decentralized strategy alignment across distributed agents while maintaining the privacy of strategy choices. To address this problem, we propose a framework that employs an energy-based model. It facilitates decentralized strategy alignment by associating agent states with coordination strategies through local minimum energy values. We address privacy concerns through a simple yet effective communication protocol that protects strategy selections from eavesdropping and information leakage. Our extensive experimental results validate these contributions, demonstrating scalability and reduced computational demands. This enhances the practicality of coordination systems in multi-agent settings.

## 1 INTRODUCTION

Multi-agent coordination has been an active research area for years (Busoniu et al., 2008; Cao et al., 2012; Yan et al., 2013; Torreno et al., 2017; Rizk et al., 2019; Gronauer & Diepold, 2022). By coordinating, agents can interact in a structured manner, optimizing the collective outcomes of their efforts. This coordination is applied across various fields, including drone swarms, computer networks, and emergency response. Effective coordination results in more robust and adaptable systems, capable of handling complex scenarios more effectively. The current study of coordination presents a variety of challenging research problems including multi-agent reinforcement learning (Lowe et al., 2017; Zhang et al., 2020; 2021), communication efficiency (Jiang & Lu, 2018; Zhang et al., 2019), uncertainty handling (Oliehoek et al., 2016; Foerster et al., 2017), task allocation (Matarić et al., 2003; Skaltsis et al., 2021), the scalability of coordination (Arslan & Yüksel, 2016; Qu et al., 2020), cooperative games (Rahwan et al., 2015; Chalkiadakis et al., 2022), *etc*. These challenges drive innovations to push the boundaries of what multi-agent systems can achieve.

While the existing research has produced fruitful results, the problem of coordination strategy alignment is almost unexplored. Specifically, in many real scenarios (*e.g*., security management for ad hoc networks, military operations, *etc*.), there can be multiple strategies to be selected. Coordination strategy alignment means all cooperative agents follow the same strategy and execute the prescribed actions. Typically, the literature implicitly assumes the presence of a global coordinator who synchronizes the strategy selection across all agents (Celli & Gatti, 2018; Farina et al., 2018; Cacciamani et al., 2021). This centralized approach, while simplifying strategy alignment, poses a high requirement on the communication infrastructure, *i.e*., the global coordinator needs to communicate with every agent for every alignment. This centralized communication often impedes scalability and is infeasible in environments without a global communication infrastructure.

Another overlooked aspect is the preservation of privacy concerning the chosen coordination strategy. The privacy of current strategy selection is important in competitive environments. Revealing the current strategic choice can significantly degrade coordination performance as it allows opponents to tailor their responses more effectively. To ensure privacy, communication channels can be secured

with encryption techniques, though this increases computational and communication resource usage. Additionally, considering potential information leakage from agents within the cooperative team introduces a complex requirement: each agent should know only its specific role as dictated by the coordination strategy, without awareness of the overall strategy.

In this paper, we propose *federated coordination*, a new research problem aiming to achieve decentralized strategy alignment while maintaining the privacy of the current strategy selection. As a first attempt to address this problem, we introduce a novel framework that employs an energy-based model to establish correlations among agents. Specifically, given a set of energy states (each held by one agent), the energy-based model returns a specific energy value. Some sets of energy states result in local minimum energy and are termed local minima. ***Our key idea is to associate each local minimum with a specific coordination strategy.*** That is, we set each agent's energy state of a local minimum as a key in the agent's dictionary, with the agent's actions prescribed by the associated strategy as the value. For each coordination alignment, every agent randomly initializes its energy state and participates in an energy minimization process. This process keeps updating the energy states until they converge to a certain local minimum. Then, each agent uses its energy state within the attained local minimum as a key, and executes the actions designated under that key. Collectively, the actions performed by all agents constitute the coordination strategy corresponding to the achieved local minimum, thereby ensuring the alignment of the coordination strategy of agents.

Crucially, our algorithm only requires local communication. During energy minimization, each agent only needs to know the energy state of its immediate neighbors to compute the gradient of its own energy state, facilitating a decentralized coordination process. This reduces the dependence on centralized communication infrastructure and facilitates the deployment in wild areas. Moreover, each agent knows only its own actions, yet the collective strategy is aligned across all agents.

Regarding privacy, we consider three levels of opponent abilities: 1) predict the coordination strategy distribution based on historical selections, 2) eavesdrop on communicated information, and 3) access to confidential information stored by some agents. We show that our distributed strategy alignment process enables the strategy distribution to be non-stationary (§ 4.3), which prevents the opponent from making accurate predictions. Moreover, we develop an encryption-free yet privacy-preserving communication protocol, ensuring that the current strategy choice remains confidential even if an opponent intercepts all communication. Furthermore, our protocol guarantees that even if some agents disclose their own information, the confidentiality of the data pertaining to agents not directly connected to any compromised agents remains protected.

To summarize, this paper makes three key contributions. 1) We introduce federated coordination, a new problem that focuses on achieving distributed strategy alignment while preserving the privacy of strategy selections. 2) We propose a novel framework to address the proposed federated coordination problem. It provably synchronizes agents' strategies without centralized control, exhibits scalability and robustness, and protects the privacy of strategy selections against eavesdropping and information leakage. 3) We conduct comprehensive experiments to validate the effectiveness of our framework and explore its various attributes, *e.g.*, scalability, robustness, and reduced computational demands.

## 2 RELATED WORK

**Multi-agent reinforcement learning (MARL)** aims to learn coordination strategies (Sunehag et al., 2017; Rashid et al., 2018; Son et al., 2019) rather than strategy alignment. It often adopts the centralized training with decentralized execution (CTDE) paradigm(Hernandez-Leal et al., 2019), where agents are trained centrally but execute strategies independently. While MARL can achieve distributed coordination, the learned strategies are typically stationary, being vulnerable to adversarial training attacks (Gleave et al., 2020). Cacciamani et al. (2021) propose to switch strategies by a global signal. This reliance on a global signal poses challenges in scenarios without a centralized coordinator and can compromise the scalability and privacy of the system.

**Consensus algorithms** are widely used to achieve distributed agreement on a common value among agents (Proskurnikov et al., 2016; Li & Tan, 2019; Amirkhani & Barshooi, 2022). However, all agents agreeing on the same value means one compromised agent can leak critical information about the whole system. In comparison, in our method, agents converge to different energy states, with these states collectively corresponding to a joint action. Our method can be viewed as a more general form of consensus, aiming to achieve a coordinated outcome instead of agreeing on a single value.

**Game theory** provides a mathematical framework for analyzing strategic interactions among agents.

Concepts such as multiple equilibria and adversarial team games are particularly relevant (von Stengel & Koller, 1997; Kalogiannis et al., 2022; Anagnostides et al., 2023). We build on these concepts by addressing the strategy alignment problem in a distributed manner, ensuring that agents can collectively reach a favorable equilibrium without revealing their strategy choices to opponents.

**Differential privacy** is a framework designed to ensure that the removal or addition of a single data point does not significantly affect the overall outcome, thus protecting individual data entries from being inferred (Dwork, 2008; Gong et al., 2020). Hence, it protects the algorithm inputs, whereas our approach focuses on maintaining the privacy of strategy selections, *i.e.*, the algorithm outputs.

In summary, while studies in MARL, consensus algorithms, and game theory are all related to coordination, they often neglect how to align agents' strategies in a distributed manner. Differential privacy mainly focuses on protecting the privacy of data inputs rather than coordinated outcomes. Our proposed framework for federated coordination addresses these gaps by enabling decentralized strategy alignment and leveraging an energy-based model for flexible and secure coordination.

## 3 PRELIMINARY

**Motivating application:** Consider a scenario where drone swarms are deployed in remote areas without access to centralized servers. These drones are equipped with various strategies, e.g., different attack formations. To optimize their performance, the drones must randomly switch between these strategies for each operation, ensuring unpredictability in their tactics Paruchuri et al. (2009); Yang et al. (2024). However, before each attack, all drones in the swarm must agree on the same randomly selected strategy. Given the lack of centralized coordination, the drones rely on local communication channels, which are vulnerable to eavesdropping by adversaries. This presents a significant challenge in ensuring secure and effective strategy coordination among the drones in the swarm.

### 3.1 FEDERATED COORDINATION PROBLEM

We use adversarial team games to model scenarios where cooperative agents face an opponent. A static, normal-form adversarial team game (von Stengel & Koller, 1997; Anagnostides et al., 2023) is defined by a tuple $\Gamma(\mathcal{N}, O, \mathcal{A}, \mathcal{B}, U)$. $\Gamma$ consists of a team of $N$ cooperative agents $\mathcal{N}$ facing an opponent $O$[1]. Each agent from $\mathcal{N}$ has a set of available actions $\mathcal{A}_i$, so that $\mathcal{A} := \prod_{i=1}^{N} \mathcal{A}_i$ denotes the joint action space of $\mathcal{N}$. Also, the opponent $O$ has a finite and nonempty set of actions $\mathcal{B}$. We denote by $\mathbf{a} = (a_1, \ldots, a_N) \in \mathcal{A}$ a joint action of $\mathcal{N}$, and $b \in \mathcal{B}$ an action of the opponent $O$. $U : \mathcal{A} \times \mathcal{B} \to \mathbb{R}$ is a utility function. The cooperative agents share the same utility represented by $U(\mathbf{a}, b)$ and the team game is assumed to be zero-sum, *i.e.*, the opponent's utility is $-U(\mathbf{a}, b)$. To maximize the utility, the team $\mathcal{N}$ normally has multiple coordination strategies[2], each maximizing utility against specific opponent actions. Let $\mathcal{S} := \{\mathbf{a}^m\}$ with cardinality as $S$ denote the set of strategies (the formal definition of $\mathcal{S}$ is in § A.4). Note that in normal-form games, a strategy is a joint action. In Markov games (Kalogiannis et al., 2022), a strategy is a joint policy $(\pi_1, \ldots, \pi_N)$ where $\pi_i$ is the policy of agent $i$ mapping a given state to a distribution over $\mathcal{A}_i$. Since this paper focuses on ensuring that agents follow the same strategy, whether it is a joint policy or a joint action is irrelevant. Henceforth, we will use a joint action $\mathbf{a}^m = (a_1^m, \ldots, a_N^m)$ to represent a strategy.

Existing works assume that a global coordinator synchronizes strategies among agents so that their joint action locates within $\mathcal{S}$. However, there are numerous scenarios where such a coordinator does not exist, *e.g.*, ad hoc networks, autonomous drone swarms, *etc*. In these scenarios, strategy alignment must be conducted in a distributed manner. Furthermore, due to the existence of an opponent, cooperative agents must preserve the privacy of their current strategy selection. Otherwise, the opponent can choose the best response $b^m$ to the disclosed strategy $\mathbf{a}^m$ such that $U(\mathbf{a}^m, b^m) \leq U(\mathbf{a}^m, b) \; \forall b \in \mathcal{B}$. This introduces a new research problem, ***federated coordination***, with the following objectives: 1) the agents in $\mathcal{N}$ coordinate their actions in a distributed manner to ensure the resulting joint action $\mathbf{a}$ belongs to $\mathcal{S}$; 2) the opponent has no information about $\mathbf{a}$.

### 3.2 ENERGY-BASED MODEL

We represent the system of cooperative agents using an undirected graph. As shown in the graph representation of Figure 1(b), each node $i$ in the graph represents an agent $i$. An edge connects

---

[1]The opponent $O$ could be a set of opposing agents. For simplicity, we regard them as one agent.

[2]Depending on applications, these strategies can be rule-based or learning-based.

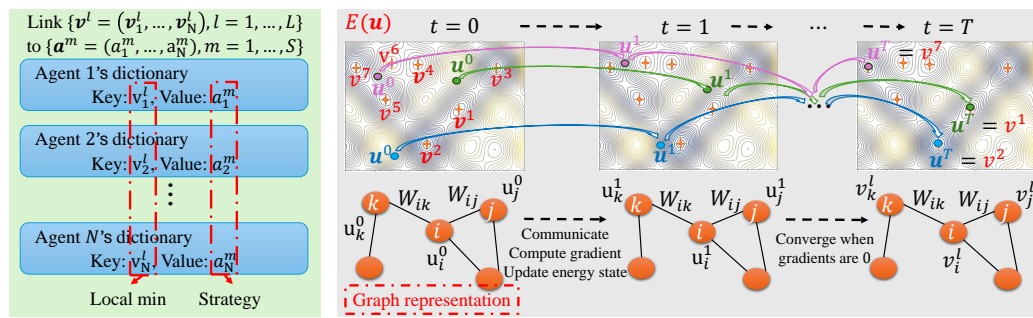

(a) Storage of coordination strategies        (b) Distributed strategy alignment

Figure 1: The proposed framework includes the storage of coordination strategies (pre-deployment) and distributed strategy alignment (post-deployment). The arrows with cyan, violet, and green color show different executions of the strategy alignment process. The local minima these executions converge to are $\mathbf{v}^2$, $\mathbf{v}^7$, and $\mathbf{v}^1$ respectively. Note that the 2-D contour map of $E(\mathbf{u})$ is only for illustrative purposes. The real contour map is high-dimensional for multiple agents.

neighboring agents that can communicate with each other. Given this graph representation, we adopt an energy-based model first studied in (Bengio & Fischer, 2015). The energy function is as follows:

$$E(\mathbf{u}) := \frac{1}{2} \sum_i \mathbf{u}_i^T \mathbf{u}_i - \frac{1}{2} \sum_{i \neq j} \rho(\mathbf{u}_i)^T \mathbf{W}_{ij} \rho(\mathbf{u}_j) - \sum_i \mathbf{b}_i^T \rho(\mathbf{u}_i) \qquad (1)$$

where $\mathbf{u}_i \in \mathbb{R}^M$ is the energy state of agent $i$, $\mathbf{b}_i \in \mathbb{R}^M$ is the energy state bias of agent $i$, $T$ means transpose operation, $\rho$ is an element-wise activation function which is $\tanh$ in this work, and $\mathbf{W}_{ij} \in \mathbb{R}^{M \times M}$ is the weight matrix of the edge connecting agent $i$ and agent $j$, and $\mathbf{W}_{ij} = \mathbf{W}_{ji}^T$. The topology of the agent network is comprehensively characterized by the adjacency matrix $\mathbf{A}$, where an entry $A_{ij} = 1$ indicates that agent $i$ is capable of communicating with agent $j$, while an entry of 0 signifies the absence of such communication, and $\mathbf{A}_{ii} = 0, \ \forall i$. In this paper, we use fixed $\mathbf{W} := \{\mathbf{W}_{ij} | A_{ij} = 1, i, j = 1, \ldots, N\}$ and $\mathbf{b} := \{\mathbf{b}_i | i = 1, \ldots, N\}$ to parameterize $E(\mathbf{u})$ while $\mathbf{u}$ are variables. Some values of $\mathbf{u}$ will achieve a local minimum value of $E(\mathbf{u})$. We let $\{\mathbf{v}^l | l = 1, \ldots, L\}$ denote the collection of these values of $\mathbf{u}$, i.e., local minima. Given $\mathbf{W}$ and $\mathbf{b}$, the size of local minimum collection $L$ and the value of each $\mathbf{v}^l$ are uniquely determined.

We choose this form of energy because it fits well with our task. In this form, the weight matrix $\mathbf{W}$ representing the connections among agents influences the energy value. Hence, a local minimum energy value establishes a correlation among agents through their connections. Moreover, the activation function ensures that the energy values are bounded and that local minima exist, which is critical for our framework. The energy state bias $\mathbf{b}$ helps adjust the value range of local minima.

## 4 PROPOSED METHOD

The proposed framework (Figure 1) consists of two components that rely on our energy-based model, i.e., the storage of coordination strategies and the distributed strategy alignment process. The storage of strategies happens before the deployment of coordination systems while the alignment process is executed every time the cooperative agents need to align their strategies after deployment.

**Storage of strategies** Given $W$ and $b$, the resulting energy-based model has multiple local minima, denoted as $\{\mathbf{v}^l = (\mathbf{v}_1^l, \ldots, \mathbf{v}_N^l) | l = 1, \ldots, L\}$. To store the set of coordination strategies $\{\mathbf{a}^m = (a_1^m, \ldots, a_N^m) | m = 1, \ldots, S\}$, we associate each of them with a certain local minimum. As depicted in Figure 1(a), we initialize one dictionary for each agent. To associate $\mathbf{a}^m$ with $\mathbf{v}^l$, we set $\mathbf{v}_i^l$ as a key and $a_i^m$ as the value in the agent $i$'s dictionary. Thereby, all strategies are stored distributively in each agent $i$ of the form $\{(\mathbf{v}_i^l : a_i^m) | l = 1, \ldots, L\}$ where ":" in the bracket represents the key-value relationship. Note that we need $L \geq S$ so that all strategies can be stored.

**Distributed alignment process** Given the storage of strategies, the distributed strategy alignment process can synchronize the agents' strategies by energy minimization. Specifically, in each execution of the alignment process, the agents first randomly initialize their energy states. Then, each agent communicates with its neighboring agents, computes the gradient of its energy state based on the received information, and updates its energy state using the gradient. Agents repeat these procedures

until the gradient is 0. At this point, the energy states will be one of the local minima $\mathbf{v}^l$. Each agent $i$ simply uses its energy state $\mathbf{v}_i^l$ to retrieve the action $a_i^m$ it needs to perform. Note that since the initial energy states are randomly generated, they will be different for different executions of the alignment process. This often results in different local minima when the alignment process converges. We demonstrate this in Figure 1(b) by putting different initial energy states $\mathbf{u}^0$ (marked with different colors) in the contour map of $E(\mathbf{u})$ and showing that they converge to different local minima.

### 4.1 Storage of coordination strategies

To associate local minima $\{\mathbf{v}^l\}$ with coordination strategies $\{\mathbf{a}^m\}$, a naive approach is to randomly generate $\mathbf{W}$ and $\mathbf{b}$, find all local minima of the resulting energy-based model, and link each local minimum to a coordination strategy. However, with random $\mathbf{W}$ and $\mathbf{b}$, determining all local minima of the corresponding energy-based model is very difficult, if not impossible. The best we can do is to repeat the following procedures: randomly initialize the energy states of all agents, conduct the energy minimization process, and identify one local minimum when the minimization process converges. Repeating these procedures is computationally expensive, and we can never be sure if we have identified all local minima. If the energy states of agents converge to an unknown local minimum during distributed strategy alignment, these states cannot be used to retrieve agents' actions since the unknown local minimum is not associated with any strategy. In addition, it is possible that the number of local minima $L$ determined by the random $\mathbf{W}$ and $\mathbf{b}$ is smaller than the number of strategies. This means some strategies will never be selected through the distributed strategy alignment, which could be undesired.

To address the above issues, we propose an algorithm that computes $\mathbf{W}$ and $\mathbf{b}$ based on preset local minima. This algorithm ensures that all local minima are known and that their number is greater than the number of strategies. Specifically, the algorithm leverages the fact that the gradient of each energy state $\mathbf{v}_i$ within a local minimum $\mathbf{v}$ is 0, *i.e.*,

$$\frac{\partial E(\mathbf{v})}{\partial \mathbf{v}_i} = \mathbf{v}_i - \rho'(\mathbf{v}_i) \odot \big[ \sum_{j \neq i} \mathbf{W}_{ij} \rho(\mathbf{v}_j) + \mathbf{b}_i \big] = 0, \tag{2}$$

where $\odot$ represents element-wise multiplication. Based on Equation 2, given a preset collection of local minima, we can establish a set of constrained equations where the local minima are known while $\mathbf{W}$ and $\mathbf{b}$ are the unknowns to be solved. In particular, let $\{\mathbf{v}^l | l = 1, \ldots, L\}$ denote the preset local minima. The set of constrained equations is expressed as follows:

$$\frac{\mathbf{v}_i^l}{\rho'(\mathbf{v}_i^l)} = \sum_{j \neq i} \mathbf{W}_{ij} \rho(\mathbf{v}_j^l) + \mathbf{b}_i, \quad l = 1, \ldots, L, \quad i = 1, \ldots, N. \tag{3}$$

To determine $\mathbf{W}$ and $\mathbf{b}$ from these linear equations, the number of constraints, $LNM$, must be sufficiently large to match or exceed the degrees of freedom in $\mathbf{W}$ and $\mathbf{b}$, which is given by $\frac{M^2}{2} \sum_{i,j} A_{ij} + NM$. This ensures that $\mathbf{W}$ and $\mathbf{b}$ are uniquely determined by the constraints.

To address Equation 3, we synthesize $N$ individual agent-based equations into one linear equation:

$$\mathbf{Y}^l = \overline{\mathbf{W}} \cdot \mathbf{X}^l + \overline{\mathbf{b}}, \tag{4}$$

where the block matrix $\overline{\mathbf{W}}$ and vectors are defined as follows:

$$\mathbf{X}^l = \begin{pmatrix} \rho(\mathbf{v}_1^l) \\ \vdots \\ \rho(\mathbf{v}_N^l) \end{pmatrix}, \quad \mathbf{Y}^l = \begin{pmatrix} \frac{\mathbf{v}_1^l}{\rho'(\mathbf{v}_1^l)} \\ \vdots \\ \frac{\mathbf{v}_N^l}{\rho'(\mathbf{v}_N^l)} \end{pmatrix}, \quad \overline{\mathbf{W}} = \begin{pmatrix} A_{11}\mathbf{W}_{11} & \cdots & A_{1N}\mathbf{W}_{1N} \\ \vdots & \ddots & \vdots \\ A_{N1}\mathbf{W}_{N1} & \cdots & A_{NN}\mathbf{W}_{NN} \end{pmatrix}, \quad \overline{\mathbf{b}} = \begin{pmatrix} \mathbf{b}_1 \\ \vdots \\ \mathbf{b}_N \end{pmatrix}. \tag{5}$$

Subsequently, we minimize the objective function using the method of ordinary least squares:

$$J(\mathbf{W}, \mathbf{b}) = \sum_{l=1}^{L} \left\| \overline{\mathbf{W}} \cdot \mathbf{X}^l + \overline{\mathbf{b}} - \mathbf{Y}^l \right\|^2. \tag{6}$$

Since $J(\mathbf{W}, \mathbf{b})$ is convex, we can solve for $\mathbf{W}, \mathbf{b}$ as the global minimum of $J$ by optimization.

**Value setting of $\mathbf{v}^l$.** Note that Equation 2 also holds for a local maximum. Therefore, to ensure that $\mathbf{v}^l$ represents a local minimum rather than a maximum, the Hessian matrix at $\mathbf{v}^l$:

$$\frac{\partial^2 E(\mathbf{v})}{\partial \mathbf{v}_i^l \partial \mathbf{v}_j^l} = \mathbf{I} \cdot \delta_{ij} - \mathbf{W}_{ij} \odot [\rho'(\mathbf{v}_i) \otimes \rho'(\mathbf{v}_j)] \tag{7}$$

should be positive definite, where $\delta_{ij}$ denotes the Kronecker delta and $\otimes$ represents the outer product. To this end, we observe that for sufficiently small gradients of $\rho(\mathbf{v}^l)$, characterized by $|\rho'(\mathbf{v}^l)| \ll \mathbf{1}$, the second term in Equation 7 serves only as a minimal perturbation to the identity matrix and its eigenvalues, which guarantee the positive definiteness of the Hessian matrix. Hence, the value of $\mathbf{v}^l$ should be set within the saturation region, *i.e.*, the region where the gradient of function value is sufficiently small, of $\rho$. In our experiments, we set entries of $\mathbf{v}^l$ as $\pm 3$ since $\rho$ is $\tanh$.

**Setting of $L$ and $M$.** To solve $\mathbf{W}$ and $\mathbf{b}$, we need $LNM \geq \frac{M^2}{2} \sum_{i,j} A_{ij} + NM$. In practical applications, we know the graph topology of agents, the number of agents $N$, and the number of strategies $S$ while $L$ and $M$ need to be set. However, $L$ and $M$ are inherently correlated as $M$ is the dimension of $\mathbf{W}$ and $\mathbf{b}$ which determines $L$. Arbitrarily setting $L$ and $M$ can result in an unsolvable Equation 6 or an $L$ smaller than $S$. To address this issue, we propose Algorithm 2 (§ A.5) which takes the graph topology of agents, $N$, and $S$ as inputs and computes $L$ and $M$ automatically.

**Spurious local minima and the solvability of Equation 4.** While the constructed energy function contains the prescribed local minima, spurious minima, i.e., points that are local minima of the energy function but do not belong to the set of prescribed minima, may exist. Moreover, given an arbitrary communication topology, the solvability of Equation 4 could be challenged. Please refer to § A.6 and § A.7 for a discussion about how our practical implementation addresses these issues.

## 4.2 DISTRIBUTED STRATEGY ALIGNMENT

Given the storage of strategies, the distributed strategy alignment process can align the agents' strategies by energy minimization. Given an initialized energy states $\mathbf{u}$, to minimize the energy $E(\mathbf{u})$, each agent $i$ needs to alter its own energy state $\mathbf{u}_i$ based on the gradient:

$$\frac{\partial E(\mathbf{u})}{\partial \mathbf{u}_i} = \mathbf{u}_i - \rho'(\mathbf{u}_i) \odot \Big[ \sum_{j \neq i} \mathbf{W}_{ij} \rho(\mathbf{u}_j) + \mathbf{b}_i \Big]. \tag{8}$$

According to Equation 8, each agent $i$ needs to know $\{\rho(\mathbf{u}_j)|j = 1, \ldots, N, \text{ and } A_{ij} = 1\}$ for gradient computation. Therefore, each agent $i$ should communicate with its neighboring agents to get this information. However, a poorly designed communication protocol may allow the opponent to obtain the knowledge of converged energy states and thus the current strategy selection.

Therefore, we propose a privacy-preserving communication protocol to address this issue. Specifically, during the storage of coordination strategies, we let each agent $i$ store $\{\mathbf{W}_{ij}|j = 1, \ldots, N, \text{ and } A_{ij} = 1\}$, *i.e.*, the agent $i$ shares $\mathbf{W}_{ij}$ with the neighboring agent $j$. Note that we do not let the agent $i$ store $\{\mathbf{W}_{jk}|j, k = 1, \ldots, N, \text{ and } j, k \neq i\}$ to enhance privacy. Moreover, we let all agents share the same random number generator $\mathcal{G} : \mathbb{R} \times \mathbb{N} \rightarrow \mathbb{R}^M$ and a seed generation function $\mathcal{F} : \mathbb{R}^{M \times M} \times \mathbb{N} \rightarrow \mathbb{R}$. Then, before the start of the $p$-th alignment process, each agent $i$ generates a new seed $sed_{ij}^p = \mathcal{F}(\mathbf{W}_{ij}, p)$ for the communication with the agent $j$. During the $t$-th energy minimization step of the $p$-th alignment process, the agent $i$ sends the agent $j$ the message $\mathbf{W}_{ij}\rho(\mathbf{u}_i^t) \oplus \sigma_{ij}$ where $\oplus$ is bitwise XOR operation and $\sigma_{ij} = \mathcal{G}(sed_{ij}^p, t)$ is a disruptive noise. Because the agent $j$ knows $\mathbf{W}_{ij}$, it can recover $\sigma_{ij}$ by $\mathcal{G}(\mathcal{F}(\mathbf{W}_{ij}, p), t)$ and remove the noise in the received message to obtain $\mathbf{W}_{ij}\rho(\mathbf{u}_i^t)$. Let $dict_i$ denote the dictionary of the agent $i$ that maps an energy state to an action and $opt_i : \mathbb{R}^M \times \mathbb{R}^M \rightarrow \mathbb{R}^M$ denote the optimizer used by the agent $i$ to update energy states based on gradients. We summarize the overall procedures as Algorithm 1.

---

**Algorithm 1** The overall procedures for the distributed strategy alignment.

**Input**: $\{\mathbf{W}_{ij}\}, \{\mathbf{b}_i\}, \{dict_i\}, \mathcal{G}, \mathcal{F}$, a small threshold $\epsilon$, and the number of alignment process $p$.
**Output**: $\mathbf{a} = \{a_i|i = 1, \ldots, N\}$.

1: Let $t = 0$ and randomly initialize $\{\mathbf{u}_i^t\}$.
2: **for** each agent $i$ **do**
3:     Generate $\{sed_{ij}^p = \mathcal{F}(\mathbf{W}_{ij}, p)|A_{ij} = 1\}$.
4: **end for**
5: **while** $\{\frac{\partial E(\mathbf{u}_i^t)}{\partial \mathbf{u}_i^t}\}$ are not all less than $\epsilon$ **do**
6:     **for** each agent $i$ **do**
7:         Send $\mathbf{W}_{ij}\rho(\mathbf{u}_i^t) \oplus \mathcal{G}(sed_{ij}^p, t)$ to each neighboring agent $j$.
8:         Remove $\mathcal{G}(sed_{ij}^p, t)$ in the message received from agent $j$ to get $\mathbf{W}_{ij}\rho(\mathbf{u}_j^t)$.
9:         Compute $\frac{\partial E(\mathbf{u}^t)}{\partial \mathbf{u}_i^t}$ based on Equation 8.
10:        Update $\mathbf{u}_i^{t+1} \leftarrow opt_i(\mathbf{u}_i^t, \frac{\partial E(\mathbf{u}^t)}{\partial \mathbf{u}_i^t})$.
11:     **end for**
12:     $t \leftarrow t + 1$.
13: **end while**
14: Output $\{a_i = dict_i(\mathbf{u}_i^t)|i = 1, \ldots, N\}$.

---

### 4.3 ANALYSIS OF PRIVACY-PRESERVING CAPABILITY

Before the analysis, we present the ***Threat Model*** outlining various levels of opponent ability:

$L1$**: Prediction of coordination strategy distribution.** The opponent predicts the distribution of the coordination strategies by analyzing the historical selections of cooperative agents.

$L2$**: Eavesdropping on communicated information.** The opponent eavesdrops on the communication channels used by the cooperative agents.

$L3$**: Access to Confidential Information.** The opponent can gain access to confidential information stored by some of the cooperative agents.

Moreover, all opponents know the procedures of the distributed strategy alignment process and have $\mathcal{G}$ and $\mathcal{F}$. In the following, we analyze the privacy-preserving capability of our framework when facing opponents with different abilities.

For an opponent with $L1$ ability, our framework inherently can use a distributed method to make the strategy distribution non-stationary, preventing the opponent from learning an accurate distribution. Specifically, given $\mathbf{W}$ and $\mathbf{b}$, the local minimum to which the energy states converge is determined by the initial energy states and the optimizer used to update energy states. Consequently, the strategy distribution depends on the distribution of initial energy states and the chosen optimizer. Thus, each agent can independently and periodically alter its optimizer and method for initializing energy states, ensuring a non-stationary strategy distribution.

When facing an opponent with $L2$ ability, our framework can guarantee the privacy of current strategy selection through the proposed communication protocol. Specifically, the opponent can eavesdrop the communicated messages $\{\mathbf{W}_{ij}\rho(\mathbf{u}_i^t) \oplus \mathcal{G}(sed_{ij}^p, t)|i, j = 1, \ldots, N, \text{ and } A_{ij} = 1\}$. As it does not know $\mathbf{W}$, it has no way to compute $\mathcal{G}(sed_{ij}^p, t)$ and cannot get $\mathbf{W}_{ij}\rho(\mathbf{u}_i^t)$. Therefore, the opponent has no information to infer the current strategy selection. In comparison, if the agent $i$ directly sends $\rho(\mathbf{u}_i^t)$, the opponent can compute $\{\mathbf{u}_i^t\}$ through $\{\rho^{-1}(\rho(\mathbf{u}_i^t))\}$. Given a converged $\{\mathbf{u}_i^t\}$, the opponent may infer the strategy selection if it has seen these converged energy states before. Alternatively, the agent $i$ may send $\mathbf{W}_{ij}\rho(\mathbf{u}_i^t)$. In this case, the opponent cannot recover $\{\mathbf{u}_i^t\}$ due to the lack of $\mathbf{W}_{ij}$. However, when the energy states converge, the opponent will obtain $\{\mathbf{W}_{ij}\rho(\mathbf{v}_i^l)\}$ corresponding to a certain $\mathbf{v}^l$. If this $\mathbf{v}^l$ has been converged to before, the opponent will find $\{\mathbf{W}_{ij}\rho(\mathbf{v}_i^l)\}$ was received before and thus can infer the current strategy as the strategy at that time. Our method prevents the above issue by adding noise $\mathcal{G}(sed_{ij}^p, t)$. It ensures that $\{\mathbf{W}_{ij}\rho(\mathbf{v}_i^l) \oplus \mathcal{G}(sed_{ij}^p, t)\}$ are different in different alignment processes even for the same $\mathbf{v}^l$.

Regarding an opponent with $L3$ ability, our framework can protect the information of the agents not directly connecting with the compromised agents. Assuming the agent $k$ in Figure 1(b) is compromised, which means the opponent knows $\mathbf{W}_{ik}$. Based on the received $\mathbf{W}_{ik}\rho(\mathbf{u}_k^t) \oplus \mathcal{G}(sed_{ik}^p, t)$ and $\mathbf{W}_{ik}\rho(\mathbf{u}_i^t) \oplus \mathcal{G}(sed_{ik}^p, t)$, the opponent can recover $\mathbf{u}_k^t$ and $\mathbf{u}_i^t$ because it can compute $\mathcal{G}(sed_{ik}^p, t)$ through $\mathbf{W}_{ik}$. However, $\mathbf{u}_j^t$ is still safe as it is included in the message $\mathbf{W}_{ij}\rho(\mathbf{u}_j^t) \oplus \mathcal{G}(sed_{ij}^p, t)$ and the opponent cannot recover it without the knowledge of $\mathbf{W}_{ij}$.

## 5 EXPERIMENTS

In the experiments, we set entries of $\mathbf{v}^l$ as $\pm 3$ to ensure each $\mathbf{v}^l$ is a local minimum (please check § 4.1 for detailed discussion). We use the optimizer *Adam* (Kingma & Ba, 2014) with learning rate as $10^{-3}$ to both solve Equation 6 for getting $\mathbf{W}$ and $\mathbf{b}$ from preset $\{\mathbf{v}^l\}$ and update energy states during the energy minimization. We set $\epsilon$ as $10^{-4}$. Networks tested with loss connections in § 5.4 are generated with initial connection probability of 0.5. For the experiments in § 5.2, § 5.3, and § 5.4, we run with 5 random seeds and show standard deviations. We run all experiments in Ubuntu 22.04 LTS system with 13th Gen Intel(R) Core(TM) i9-13900KF CPU and Nvidia 4090 GPU.

### 5.1 EFFECTIVENESS

In this experiment, we verify the effectiveness of the proposed framework by assessing whether the distributed alignment process can make agents converge to a preset local minimum and whether different executions of the alignment process will converge to different local minima. We run experiments for different topologies with the number of agents being 10, 20, 40, and 80, respectively.

Moreover, for each preset local minimum $\mathbf{v}^l$, we let all its elements have the same value, *i.e.*, given an $l$, $\mathbf{v}_i^l = \mathbf{v}_j^l$ for all $i, j = 1, \ldots, N$. This setting simplifies checking whether agents converge to a

Table 1: Topologies of random networks and corresponding equilibria. The three rows show the agent number, the topological structure of agents, and the equilibria (*i.e.*, converged energy states).

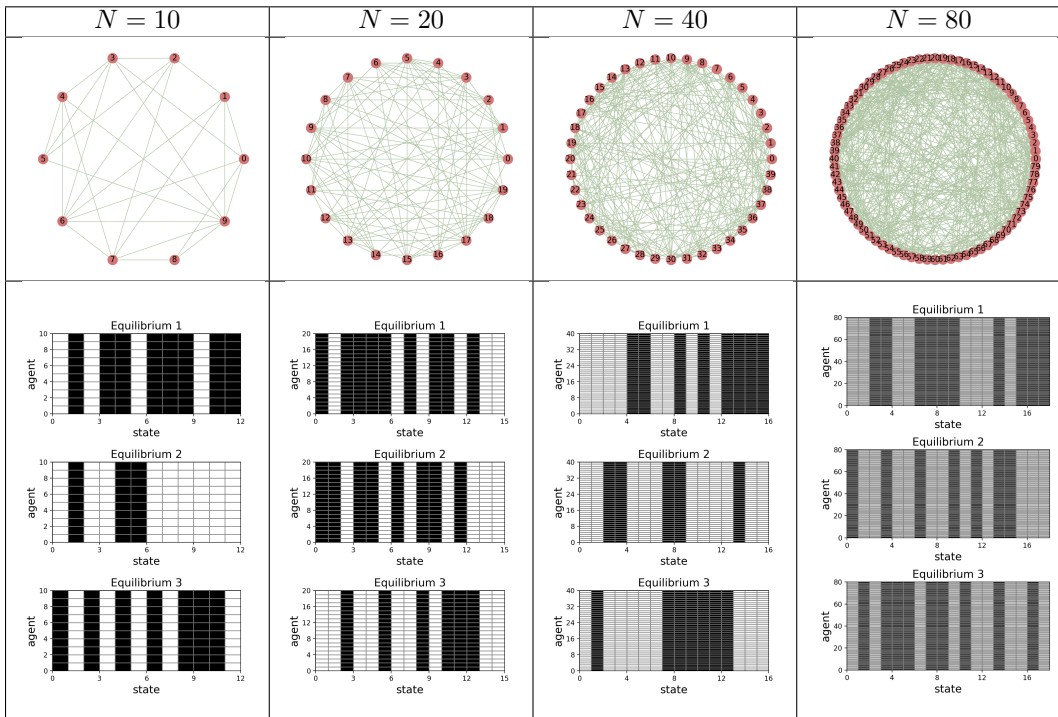

preset local minimum. That is, each entry of $\mathbf{v}_i^l$ is either 3 or $-3$, allowing us to represent $\mathbf{v}_i^l$ as a vector with binary entries. Below, we plot each entry of an agent's energy state with a black or white grid based on whether its value is closer to 3 or $-3$. Then, when we plot each agent's energy state, the distribution of black and white grids will be identical among agents if they converge to a preset local minimum because the preset minimum's elements are the same. Otherwise, the distributions will not match, which can be identified visually. Note that in practical application, we will randomly permute the elements of $\mathbf{v}^l$ to prevent the entries of $\{\mathbf{v}_i^l | i = 1, \ldots, N\}$ from being correlated.

We present our results in Table 1, 2, and 5. For each topology, we run the strategy alignment process three times. From the third row, we observe that the distributions of black and white grids are the same across agents for each obtained equilibrium (*i.e.*, the converged energy states), which means the equilibrium is a preset local minimum. Moreover, different equilibria demonstrate different distributions of black and white grids, indicating that different executions of the alignment process converge to different local minima. These results verify the effectiveness of the proposed framework.

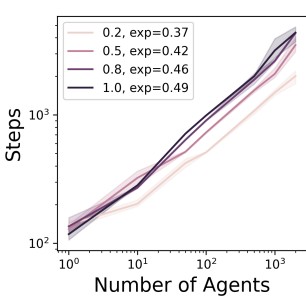

Figure 2: Log-log plot of energy minimization steps with varying connection probabilities $(0.2, 0.5, 0.8, 1.0)$. The line slopes, denoted by 'exp', indicate sub-linear complexity in the iteration steps.

## 5.2 SCALABILITY

In this section, we study the scalability of our framework, *i.e.*, how the number of steps required for energy minimization, denoted as $Z$, grows as the number of agents $N$ increases. In this experiment, given $N$, we randomly generate the topology of agents by using different connection probabilities $(0.2, 0.5, 0.8, 1.0)$ to connect each pair of agents. For each topology, we measure $Z$. We demonstrate the results in a log-log plot as shown in Figure 2 which represents the relationship between $Z$ and $N$ as $\log Z = \alpha \log N + C$ where $\alpha$ is the line slope and $C$ is a constant.

Figure 2 indicates that $\alpha$ ranges from 0.37 to 0.49, which means $Z$ increases slower than $\sqrt{N}$ (when $\alpha = 0.5$). For instance, when $N$ increases by 100 times, $Z$ increases less than 10 times.

Table 2: Topologies of small world networks and corresponding equilibria. The three rows show the agent number, the topological structure of agents, and the equilibria (*i.e.*, converged energy states).

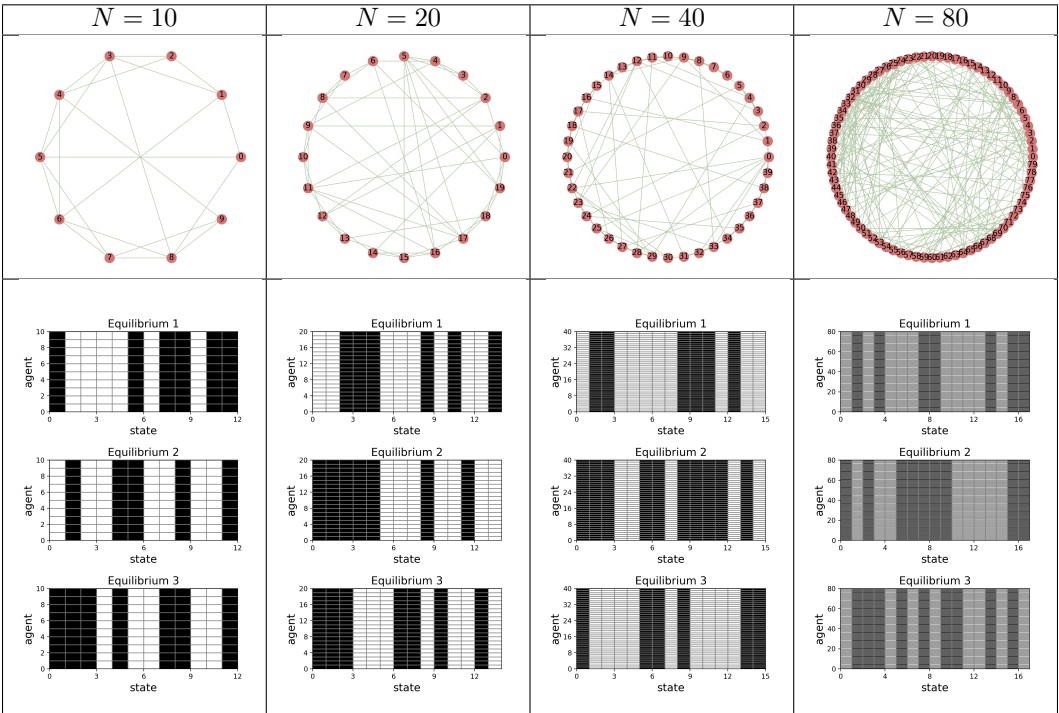

This sub-linear growth is significant as it shows that the algorithm remains efficient even as the network size expands, thus suitable for large-scale systems. Notably, $Z$ increases faster with higher connection probabilities, suggesting that more interconnected networks require additional iterations to achieve local minima. This finding highlights the trade-off between connectivity and speed of convergence and underscores the importance of optimizing network topology to balance efficiency and performance.

### 5.3 COMPUTATIONAL EFFICIENCY

The communication protocol in our proposed framework is encryption-free yet privacy-preserving. One may wonder whether we can use encryption techniques to protect the privacy of $\{\rho(\mathbf{u}_i)\}$ instead of using the noise information produced by random number generators. In this part, we compare the computational time of our method with that of the Advanced Encryption Standard (AES) which is one of the most efficient symmetric block cipher (Mahajan & Sachdeva, 2013). We use a 128-bit key size, which is widely adopted to balance security and efficiency.

Note that when using AES to protect $\rho(\mathbf{u}_i)$, it first needs to encrypt $\rho(\mathbf{u}_i)$ on the sender side and then decrypt the message on the receiver side. In contrast, our method requires performing matrix multiplication, *i.e.*, $\mathbf{W}_{ij}\rho(\mathbf{u}_i)$, and a bit-wise XOR operation on the sender side, and only a bit-wise XOR operation on the receiver side, as the receiver can directly use $\mathbf{W}_{ij}\rho(\mathbf{u}_i)$ to compute gradients. Given that the computational time of the XOR operation is negligible, we only compare the times of AES encryption (denoted as "AES Enc"), AES decryption (denoted as "AES Dec"), and the sending operation of our method (denoted as EFC, *i.e.*, energy-based federated coordination). Table 3 displays the results for handling $\rho(\mathbf{u}_i)$ of different dimensions ($M = 20, 60, 100, 140, 180$). It demonstrates that the computational time of our method is an order of magnitude lower than that of AES, indicating the computational efficiency of our method.

### 5.4 ROBUSTNESS AGAINST LOST CONNECTIONS

In real-world scenarios, the communication between agents might be unreliable or intermittent. Hence, the robustness of our framework against lost communication links is crucial for its applicability. In this experiment, the communication link between the agent $i$ and the agent $j$ is lost means $\mathbf{W}_{ij}$ becomes $\mathbf{0}$.

Table 3: Computational efficiency comparison: Encryption Time (ns).

| Method | Data Size $M$ | | | | |
|---|---|---|---|---|---|
| | 20 | 60 | 100 | 140 | 180 |
| AES Enc | $81.02 \pm 0.32$ | $82.12 \pm 0.95$ | $85.57 \pm 1.45$ | $86.21 \pm 0.39$ | $86.45 \pm 0.67$ |
| AES Dec | $71.41 \pm 0.39$ | $73.03 \pm 0.82$ | $75.78 \pm 1.49$ | $76.88 \pm 0.16$ | $77.28 \pm 0.43$ |
| EFC | $1.57 \pm 0.06$ | $1.81 \pm 0.05$ | $2.09 \pm 0.04$ | $2.91 \pm 0.03$ | $3.51 \pm 0.03$ |

We evaluate the robustness under different numbers of agents and different loss rates of communication links. That is, given $N$ and a loss rate, we first randomly generate a topology of agents and compute the corresponding $\mathbf{W}$ and $\mathbf{b}$. Next, we set each $\mathbf{W}_{ij}$ to $\mathbf{0}$ with the probability specified by the loss rate. Then, we measure the alignment success rate based on the percentage of agents that achieve a preset local minimum.

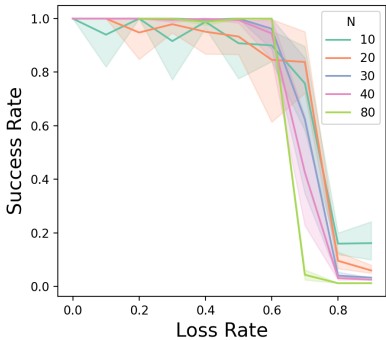

Figure 3: Success rate of achieving preset equilibria versus communication loss rate for varying numbers of agents ($N = 10, 20, 30, 40, 80$). This shows robustness in multi-agent networks.

Results in Figure 3 show that the framework upholds a high success rate up to a communication loss rate of $0.6$, signifying resilience to moderate disruptions. Beyond this point, there is a sharp decline in the success rate, which primarily arises due to the network fragmentation caused by high communication loss. That is, when communication links are lost at higher rates, agents become isolated into separate groups, hindering their ability to reach alignment and achieve a local minimum collectively. In addition, networks with more agents show greater stability and resilience against increased loss rate before the loss rate exceeds $0.6$.

## 6    DISCUSSION ON LIMITATIONS AND FUTURE WORK

As the first work addressing the challenging problem of federated coordination, our primary goal is to establish a baseline framework and demonstrate its effectiveness. However, we recognize certain limitations in our current approach and identify several promising directions for future research.

**Pre-determined mapping between energy states and strategies**    To simplify the setup and provide a clear proof of concept, our current implementation uses a pre-determined mapping between energy states and strategies. While effective for validating the framework, this approach may limit flexibility and adaptability in complex scenarios. Future research could explore dynamic and learned mappings, enabling agents to autonomously adapt energy-strategy relationships to evolving environments and coordination challenges. energy-based model could incorporate temporal dependencies to handle dynamic environments, potentially leveraging time-evolving graph structures for inter-agent interactions.

**Pre-defined strategies**    Our approach differs intentionally from MARL by focusing on decentralized alignment for pre-defined strategies. Integrating strategy learning mechanisms into our framework represents an exciting avenue for future work.

## 7    CONCLUSIONS

In this paper, we introduce federated coordination, a new problem for decentralized strategy alignment in multi-agent systems that preserves privacy. Using an energy-based model, our novel framework synchronizes agents' strategies without a central coordinator, reducing dependence on global communication and maintaining privacy through an encryption-free protocol against eavesdropping and information leakage. Extensive experiments show our framework's efficiency, scalability, lower computational demands compared to AES cipher, and robustness against communication link loss. This makes it a promising solution for decentralized, privacy-preserving multi-agent coordination.

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

## A  APPENDIX

### A.1  NOTATION TABLE

Table 4: Notation Table

| Symbol | Definition |
|---|---|
| $N$ | Number of cooperative agents in the system. |
| $\mathcal{A}$ | Joint action space of all agents. |
| $\mathcal{B}$ | Set of actions available to the adversary. |
| $U(a,b)$ | Utility function, where $a \in \mathcal{A}$ and $b \in \mathcal{B}$. |
| $E(\mathbf{u})$ | Energy function used to represent the system of agents. |
| $\mathbf{u}_i$ | Energy state of agent $i$. |
| $\mathbf{v}^l$ | Local minimum of the energy function. |
| $\rho(x)$ | Activation function (tanh in this work). |
| $\mathbf{W}_{ij}$ | Weight matrix for the edge connecting agent $i$ and agent $j$. |
| $\mathbf{b}_i$ | Energy state bias of agent $i$. |
| $\epsilon$ | Convergence threshold for energy minimization. |
| $\delta_{ij}$ | Kronecker delta, equal to 1 if $i = j$, otherwise 0. |
| $\mathcal{G}(s,t)$ | Random noise generator function with seed $s$ and step $t$. |
| $\mathcal{F}(W,p)$ | Seed generation function based on weight $W$ and alignment process $p$. |
| $S$ | Set of predefined coordination strategies. |
| $L$ | Number of local minima in the energy landscape. |
| $M$ | Dimensionality of the energy state vector $\mathbf{u}_i$. |
| $\otimes$ | Outer product operator. |
| $\oplus$ | Bitwise XOR operation. |
| $\nabla E(\mathbf{u})$ | Gradient of the energy function with respect to the energy states $\mathbf{u}$. |

### A.2  MORE DISCUSSION ON RELATED WORK

The proposed federated coordination problem encompasses several essential ingredients, including coordination, decentralized agreement, the existence of multiple strategies, and privacy. In this section, we discuss the existing research areas that study each of these aspects.

**Multi-agent reinforcement learning (MARL)** aims to learn strategies for multiple agents to facilitate their coordination(Sunehag et al., 2017; Rashid et al., 2018; Son et al., 2019). It often adopts the centralized training with decentralized execution (CTDE) paradigm(Oliehoek et al., 2008; Hernandez-Leal et al., 2019), where agents are trained centrally but execute strategies independently. While MARL can achieve distributed coordination, the primary focus is on learning coordination strategies rather than strategy alignment. Moreover, the learned strategies are typically stationary, making them vulnerable to adversarial training attacks (Gleave et al., 2020). To enable agents to switch their strategies dynamically, existing work often relies on a global signal to coordinate policy changes (Cacciamani et al., 2021). This reliance on a global signal poses challenges in scenarios without a centralized coordinator and can compromise the scalability and privacy of the system.

**Consensus algorithms** are widely used to achieve distributed agreement on a common value among agents in a network (Proskurnikov et al., 2016; Li & Tan, 2019; Amirkhani & Barshooi, 2022). These algorithms ensure that all agents eventually converge to the same value, facilitating coordinated actions. However, all agents agreeing on the same value means one compromised agent can leak critical information about the whole system. In comparison, in our proposed method, agents converge to different energy states, with these states collectively corresponding to a joint action. This approach can be viewed as a more general form of consensus, where the goal is not to agree on a single value but to achieve a coordinated outcome through distributed strategy alignment.

**Game theory** provides a mathematical framework for analyzing strategic interactions among rational agents. In the context of multi-agent systems, concepts such as multiple equilibria and adversarial team games are particularly relevant (von Stengel & Koller, 1997; Kalogiannis et al., 2022; Anagnostides et al., 2023). Our work builds on these concepts by addressing the strategy alignment problem

Table 5: Topologies of rings and corresponding equilibria. The three rows show the agent number, the topological structure of agents, and the equilibria (*i.e.*, converged energy states).

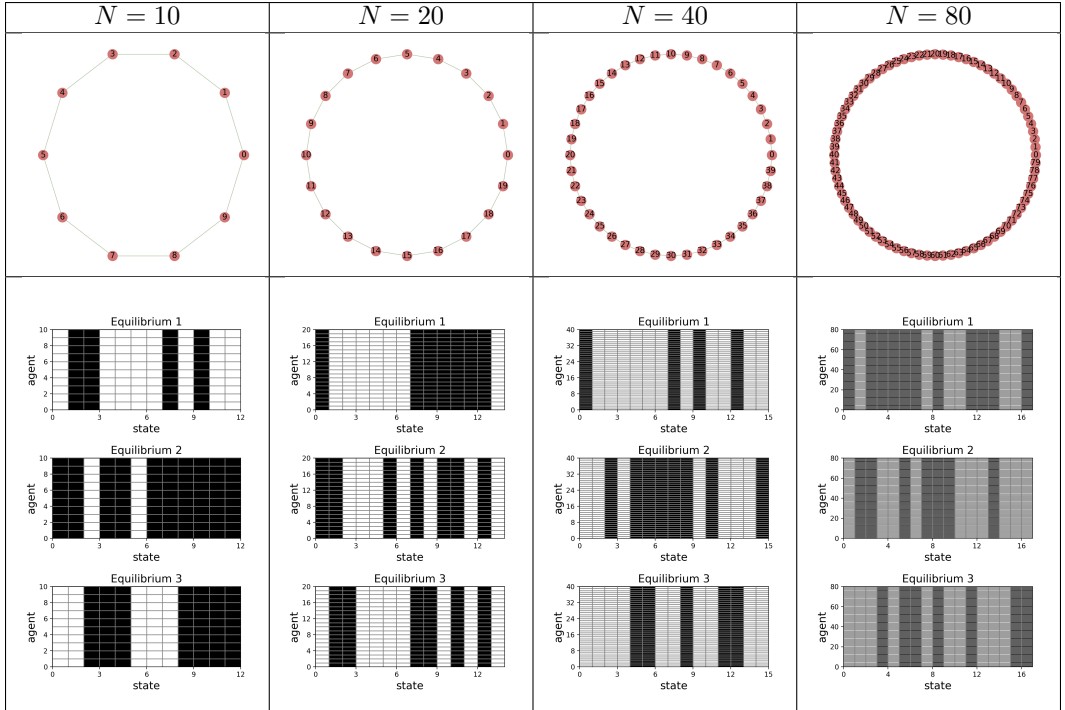

in a distributed manner, ensuring that agents can collectively reach a favorable equilibrium without revealing their strategy choices to opponents.

**Differential privacy** is a framework designed to provide privacy guarantees when analyzing and sharing statistical data (Dwork, 2008; Gong et al., 2020). It ensures that the removal or addition of a single data point does not significantly affect the overall outcome, thus protecting individual data entries from being inferred. Both differential privacy and our proposed framework aim to protect sensitive information from opponents. However, the objects under protection differ. Differential privacy applies noise to obscure individual data, i.e., the algorithm inputs, whereas our approach focuses on maintaining the privacy of strategy selections, *i.e.*, the algorithm outputs.

In summary, while studies in MARL, consensus algorithms, and game theory are all related to coordination, they often neglect how to align agents' strategies in a distributed manner. Differential privacy mainly focuses on protecting the privacy of data inputs instead of the coordinated outcomes. Our proposed framework for federated coordination addresses these gaps by enabling decentralized strategy alignment and leveraging an energy-based model for flexible and secure coordination, which enhances the practicality of coordination systems.

These distinctions highlight that federated coordination is not merely an extension of decentralized coordination or consensus but rather a novel framework designed specifically for adversarial multi-agent environments. It introduces privacy as a critical dimension, alongside decentralized strategy alignment, to address challenges that are not effectively handled by existing methods.

### A.3  ADDITIONAL RESULTS OF EFFECTIVENESS EXPERIMENT

### A.4  FORMAL DEFINITION OF COORDINATION STRATEGY SET

A static, normal-form adversarial team game (von Stengel & Koller, 1997; Anagnostides et al., 2023) is defined by a tuple $\Gamma(\mathcal{N}, O, \mathcal{A}, \mathcal{B}, U)$. $\Gamma$ consists of a team of $N$ cooperative agents $\mathcal{N}$ facing an opponent $O$. Each agent from $\mathcal{N}$ has a set of available actions $\mathcal{A}_i$, so that $\mathcal{A} := \prod_{i=1}^{N} \mathcal{A}_i$ denotes the joint action space of $\mathcal{N}$. Also, the opponent $O$ has a finite and nonempty set of actions $\mathcal{B}$. We denote by $\mathbf{a} = (a_i, \ldots, a_N) \in \mathcal{A}$ a joint action of $\mathcal{N}$, and $b \in \mathcal{B}$ an action of the opponent $O$.

$U : \mathcal{A} \times \mathcal{B} \to \mathbb{R}$ is a utility function. The cooperative agents share the same utility represented by $U(\mathbf{a}, b)$ and the team game is assumed to be zero-sum, *i.e.*, the opponent's utility is $-U(\mathbf{a}, b)$.

Below, we formally define the set of coordination strategies from which cooperative agents choose.

**Coordination strategy set** Given two joint actions $\mathbf{a}^m$ and $\mathbf{a}^n$ of $\mathcal{N}$, we say $\mathbf{a}^m$ is *strictly better* than $\mathbf{a}^n$ if $U(\mathbf{a}^m, b) > U(\mathbf{a}^n, b)$, $\forall b \in \mathcal{B}$. A joint action $\mathbf{a}^m \in \mathcal{A}$ is optimal with respect to an opponent action $b \in \mathcal{B}$ if $U(\mathbf{a}^m, b) \geq U(\mathbf{a}^n, b)$, $\forall \mathbf{a}^n \in \mathcal{A}$. Let $\gg$ denote the strictly better relationship and $\mathbf{a} \xrightarrow{*} b$ denote the joint action $\mathbf{a}$ being optimal with respect to the opponent action $b$. The set of coordination strategies $\mathcal{S} := \{\mathbf{a}^m \in \mathcal{A} \mid \exists b \in \mathcal{B}, \mathbf{a}^m \xrightarrow{*} b$ and $\forall \mathbf{a}^n \in \mathcal{A} \setminus \mathcal{S}, \exists b \in \mathcal{B}$ such that $\mathbf{a}^n \xrightarrow{*} b \implies \exists \mathbf{a}^m \in S$ such that $\mathbf{a}^m \gg \mathbf{a}^n\}$. This means that for every $\mathbf{a}^m \in \mathcal{S}$, it is optimal with respect to a certain opponent action $b \in \mathcal{B}$. Furthermore, for any joint action $\mathbf{a}^n \notin \mathcal{S}$ that is optimal for a certain opponent action, there exists a joint action $\mathbf{a}^m$ in $\mathcal{S}$ that is strictly better than $\mathbf{a}^n$. It implies that for any joint action $\mathbf{a}^n \notin \mathcal{S}$, it is either not optimal for any opponent actions or there exists at least a joint action $\mathbf{a}^m \in \mathcal{S}$ that is strictly better than it. We can construct $\mathcal{S}$ by first grouping all joint actions that are optimal for certain opponent actions and then removing any joint actions for which a strictly better alternative exists.

### A.5 THE ALGORITHM FOR SETTING $L$ AND $M$

Given an adjacency matrix $\mathbf{A}$ and the number of strategies $S$, our objective is to determine the number of local minima $L$ and the state size $M$. For parameterization, the degrees of freedom for $\mathbf{W}$ and $\mathbf{b}$ are given by $\frac{M^2}{2} \sum_{i,j} A_{ij}$ and $NM$, respectively. On the constraint side, the number of constraints is $LMN$. The difference between the degrees of freedom and the constraints can be denoted as $R(M, L)$:

$$R(M, L) = NML - \frac{M^2}{2} \sum_{i,j} A_{ij} - NM.$$

Two constraints are under consideration: (1) the number of constraints must be equal to or exceed the degrees of freedom, i.e., $R(M, L) \geq 0$, and (2) the number of minima must be equal to or exceed the number of strategies, i.e., $L \geq S$. Our goal is to find the minimum values of $M$ and $L$ such that these constraints are satisfied.

In binary coding, the maximum number of minima is given by $2^M$, where each state dimension corresponds to an independent local minimum dimension. However, in general, we can make $\xi$ ($0 \leq \xi \leq M$) entries of the state invariant among the minima, resulting in the number of minima $L_\xi(M) = 2^{M-\xi}$. By substituting $L$ in $R(M, L)$ with $L_\xi(M)$, we obtain:

$$R_\xi(M) = NM \cdot 2^{M-\xi} - \frac{M^2}{2} \sum_{i,j} A_{ij} - NM.$$

For $M \in \mathbb{Z}$, we seek the smallest $M$ that satisfies constraint (1), defined as $M^* = \inf\{M \in \mathbb{Z} \mid R_\xi(M) \geq 0\}$. To meet constraint (2), we may find an appropriate $M^*$ such that $L_\xi(M^*) \geq S$ by adjusting $\xi$. The following proposition will be instrumental in this process.

**Proposition 1.** *The number of minima $L_\xi(M^*)$, evaluated at $M^* = \inf\{M \in \mathbb{Z} \mid R_\xi(M) \geq 0\}$, increases monotonically with $\xi$ for $M^* \geq 2$.*

*Proof.* We start by examining the root of $R_\xi(M)$:

$$R_\xi(M) = NM \cdot 2^{M-\xi} - \frac{M^2}{2} \sum_{i,j} A_{ij} - NM = 0$$

Dividing by $NM$, we get:

$$2^{M-\xi} - \frac{M}{2N} \sum_{i,j} A_{ij} - N = 0.$$

Define $F(M, \xi) = 2^{M-\xi} - (aM + b)$ where $a = \frac{1}{2N} \sum_{i,j} A_{ij}$ and $b = N$. We seek $M$ such that $F(M, \xi) = 0$. To understand how $M$ changes with $\xi$, we differentiate $F(M, \xi) = 0$ implicitly with respect to $\xi$:

$$\frac{dF}{d\xi} = \frac{\partial F}{\partial M} \frac{dM}{d\xi} + \frac{\partial F}{\partial \xi} = 0.$$

We compute the partial derivatives:

$$\frac{\partial F}{\partial M} = \frac{\partial}{\partial M}(2^{M-\xi} - aM - b) = 2^{M-\xi} \ln 2 - a,$$

$$\frac{\partial F}{\partial \xi} = \frac{\partial}{\partial \xi}(2^{M-\xi} - aM - b) = -2^{M-\xi} \ln 2.$$

Substituting these into the implicit differentiation equation, we obtain:

$$(2^{M-\xi} \ln 2 - a)\frac{dM}{d\xi} - 2^{M-\xi} \ln 2 = 0.$$

Solving for $\frac{dM}{d\xi}$, we get:

$$\frac{dM}{d\xi} = \frac{2^{M-\xi} \ln 2}{2^{M-\xi} \ln 2 - a}.$$

We now differentiate $L_\xi(M) = 2^{M-\xi}$ with respect to $\xi$:

$$\frac{d}{d\xi}(2^{M-\xi}) = 2^{M-\xi} \ln 2 \left( \frac{dM}{d\xi} - 1 \right) = \frac{2^{M-\xi} \ln 2 \cdot a}{2^{M-\xi} \ln 2 - a}.$$

For $M$ as the root of $R_\xi(M)$, we can replace $2^{M-\xi}$ with $aM + b$, thus:

$$\frac{d}{d\xi}(2^{M-\xi}) = \frac{2^{M-\xi} \ln 2 \cdot a}{(aM + b) \ln 2 - a}, \tag{9}$$

which is positive for $M > \frac{1}{\ln 2} \approx 1.44$. For $M^* = \inf\{M \in \mathbb{Z} \mid R_\xi(M) \geq 0\}$, it is given as $M^* = \lceil M \rceil$, and the positiveness of Equation 9 is still maintained for $M^* \geq 2$, which means $L_\xi(M^*)$ increases monotonically with $\xi$ for $M^* \geq 2$ □

Therefore, we may increment $\xi$ and solve for $M^*$ until both constraints are satisfied. The process of determining $\xi$ and the required $M$ and $L$ is detailed below in Algorithm 2.

### A.6 ADDRESSING THE POSSIBILITY OF SPURIOUS MINIMA

While our proposed approach ensures that the prescribed local minima correspond to those of the constructed energy function, it does not theoretically eliminate the possibility of spurious minima—points that are local minima of the energy function but do not belong to the set of prescribed minima. This subsection discusses the implications of spurious minima and the practical strategies employed in our implementation to address this issue.

---

**Algorithm 2** The algorithm for setting $L$ and $M$ automatically

---

**Input**: $\mathbf{A} \in \{0,1\}^{N \times N}$.
**Output**: $M, L \in \mathbb{Z}$.
1: $\xi = 0$.
2: def $L(M, \xi) = 2^{M-\xi}$
3: def $R(M, \xi) = NM \cdot L(M, \xi) - \frac{M^2}{2} \sum_{i,j} A_{ij} - NM$.
4: solve for $M^* = \inf\{M \in \mathbb{Z} | R(M, \xi) \geq 0\}$
5: **while** $L(M^*, \xi) < S$ **do**
6:     $\xi$ += 1
7:     solve for $M^* = \inf\{M \in \mathbb{Z} | R(M, \xi) \geq 0\}$
8: **end while**
9: Output $M^*, L(M^*)$.

---

A.6.1    THEORETICAL CONSIDERATIONS

The energy function constructed in our framework is designed to ensure that the prescribed local minima satisfy the optimization constraints in Equation 6. However, due to the complexity of the energy landscape, it is theoretically possible for spurious minima to exist. These spurious minima may not align with the prescribed strategy set, potentially affecting the robustness of strategy alignment.

A.6.2    MITIGATION THROUGH BINARY CODING

To address the potential presence of spurious minima, our implementation uses **binary coding**, as described in § A.5, to construct the energy function and map spurious minima to prescribed ones:

- The number of prescribed local minima, $L$, is determined as $2^M$, where $M$ is the dimension of the energy states.
- Algorithm 2 identifies the minimum $M$ and a parameter $\xi$ to ensure:
    1. $2^{M-\xi} NM \geq \frac{M^2}{2} \sum_{i,j} A_{ij} + NM$, where the left-hand side represents the number of constraints, and the right-hand side represents the degrees of freedom in $\mathbf{W}$ and $\mathbf{b}$.
    2. $2^{M-\xi} \geq S$, where $S$ is the number of strategies.

After determining $M$ and $\xi$, we construct $2^{M-\xi}$ local minima with:

- The first $M - \xi$ entries of each state set to either $3$ or $-3$.
- The remaining $\xi$ entries fixed at $3$.

During strategy alignment, the final converged states are determined based on whether each entry is closer to $3$ or $-3$. This mapping effectively aligns any spurious minima with the prescribed ones, ensuring that agents align on a certain coordination strategy.

We conducted extensive experiments to validate the robustness of our implementation against spurious minima. These experiments include several detailed tests for various topologies (as outlined in Tables 1, 2, and 5) and thousands of additional trials. In all cases, the converged states aligned with the predetermined minima, demonstrating the effectiveness of our implementation.

A.7    DISCUSSION ON SOLVABILITY OF EQUATION 4

The existence of solutions to Equation 4 is mathematically intricate due to implicit constraints imposed by the topology $A$ and the predetermined set of local minima $\{\mathbf{v}^l\}$.

The block matrix $\overline{\mathbf{W}}$ defined in Equation 5 is subject to implicit constraints determined by the topology $A$. To elucidate these constraints, consider a scenario where the inter-agent weights are homogeneous, i.e., $\mathbf{W}_{ij} = \mathbf{W}_{i'j'}$ for all indices $i, j, i', j'$. In this case, the block matrix can be represented as a Kronecker product, $\overline{\mathbf{W}} = A \otimes \mathbf{W}$, implying that $\det(\overline{\mathbf{W}}) = \det(A) \det(\mathbf{W})$. If the communication network is singular, indicated by $\det(A) = 0$, then the block matrix is inherently

singular, resulting in $\det(\overline{\mathbf{W}}) = 0$. Consequently, Equation 4 becomes unsolvable. In more general cases as outlined in Equation 5, the relationship involving $\overline{\mathbf{W}}$ becomes more complex. There is no straightforward method to address solvability in the most general form, thus an optimization-based approach is employed, utilizing $J(\mathbf{W}, \mathbf{b})$ to assess the problem's solvability.

In this approach, $\overline{\overline{\mathbf{W}}}$ is treated as a parameter within a linear system. The solution to Equation (4) can fall into one of three categories: (1) infinitely many solutions, (2) a unique solution, or (3) no solution, contingent on the relationship between the degrees of freedom (d.o.f) and the number of constraints. In both cases (1) and (2), the optimal value of $J^*(\mathbf{W}, \mathbf{b})$ is 0, while in case (3), $J^*(\mathbf{W}, \mathbf{b})$ is greater than 0.

Case (1): To eliminate this scenario, Algorithm 2 is employed to ensure that the number of constraints is at least equal to the d.o.f. Subsequently, a small Gaussian noise (standard deviation$< 0.0001$) is added to each preset binary-coded local minimum. This guarantees the linear independence of the equations specified by Equation 3, thereby eliminating case (1).

Case (2): This is the desired outcome. Adding Gaussian noises to the preset local minima results in $\mathbf{W}$ and $\mathbf{b}$ that cause the real local minima to deviate slightly from the preset values, e.g., a preset minimum of 3 may result in real minima of 3.00005, 3.0001, etc. However, since the converged state is mapped to binary encoding, this ensures alignment with the preset local minima.

Case (3): In this instance, an optimal solution, denoted as $\mathbf{W}^*$ and $\mathbf{b}^*$, can still be identified by minimizing the least-squares error $J(\mathbf{W}, \mathbf{b})$. This solution does not exactly satisfy Equation (4), and $J(\mathbf{W}^*, \mathbf{b}^*)$ quantifies the deviation of $\{\mathbf{v}^l\}$ from the true local minima determined by $\mathbf{W}^*$ and $\mathbf{b}^*$. Given the convexity of $J$, the loss is distributed among $\{\mathbf{v}^l\}$. A small $J(\mathbf{W}^*, \mathbf{b}^*)$ (average loss per local minimum $< 0.1$) indicates proximity of true local minima to each preset $\mathbf{v}^l$. The binary coding mapping ensures that the true minimum aligns with the preset minimum. Conversely, a large $J(\mathbf{W}^*, \mathbf{b}^*)$ suggests that the current communication topology is inadequate for achieving alignment, prompting adjustments to the topology prior to system deployment.

