# OpenReview forum: "Federated Coordination: Private and Distributed Strategy Alignment"
_ICLR.cc/2025/Conference — Submitted to ICLR 2025_

### Official Review · Reviewer_Sc7y · 2024-11-04

**Soundness:** 3
**Presentation:** 3
**Contribution:** 3
**Rating:** 6
**Confidence:** 3

**Summary:**

The paper addresses the problem of coordination of agents without a central coordinator, addressing specifically the issues of scalability and privacy. It proposes a framework that uses energy as the base to accomplish federated coordination.  The basic idea is that each agent needs to minimize its energy use.  Each agent needs to know the energy state of its neighbors to be able to compute the gradient of its own energy.  Communication is limited to communication with neighbors., which is needed to compute the energy gradient.  For privacy, reasons, each agent knows only its own actions, not the collective strategy. To guarantee privacy, the paper considers the ability of the other agents to (1) predict the coordination strategy using past history, (2) eavesdrop om communication, and (3) access information stored by other agents. The approach proposed uses a non-stationary strategy distribution which is hard to predict, a privacy-preserving communication protocol that guarantees the confidentiality of data of agents not directly connected to compromised agents, so they stay protected. The experimental results include experiments to measure the effectiveness of the method, scalability, computational efficiency, and robustness against lost connections.

**Strengths:**

The approach proposed is a novel way of dealing with the important problem of distributed decision making with guarantees on privacy and scalability.

Since the strategy distribution depends on the distribution of the initial energy states and the optimizer, each agent can make its strategy distribution non stationary so the opponents cannot make accurate predictions.

The set of experiments included in the paper is extensive, and shows strong scalability, computational efficiency, and robustness against dropped connections.

**Weaknesses:**

The writing is clear but dense, so it is hard to follow the mathematical details.  For instance, the explanations for Tables 1, 2, 3, are not clear.  The explanation says "From the third row, we observe that the distributions of black and white grids are the same across agents for each obtained equilibrium (i.e., the converged energy states), which means the equilibrium is a preset local minimum." If I look at the table, for each number of agents there are three different equilibria. Are you saying that all the agents for that network have the same equilibria? The figures do not show that the equilibria shown are for all the agents.
Please provide a more detailed explanation of Tables 1, 2, and 3, perhaps with a step-by-step breakdown of how to interpret the data. Additionally, please clarify whether the equilibria shown are indeed for all agents in each network.

**Questions:**

What happens if there are no neighbors within the communication range?  Please discuss the implications of this scenario on your framework's performance or explain how your method handles such edge cases

Since all the agents share the same random number generator, doesn't this provide a potential path for breaking privacy? Please discuss the security implications of using a shared random number generator and whether you have considered alternative approaches to address this potential issue.

The paper says that opponents cannot infer the strategy from the messages exchanged, because noise is added to avoid cases where the opponent  has seen the converged energy states before. I am wondering if can comment on how robust this noise addition is to a tenacious set of opponents. Please provide more details on the noise addition process and discuss any theoretical or empirical guarantees you can offer regarding its robustness against persistent adversaries.

What is the time unit in Table 4? Please clarify the time unit used in Table 4 and explain how this unit relates to real-world performance

Is there a reason for Figure 3 to get up to 40 agents, while the other go up to 80 or 180?  How sharp is the decline in the success rate? Please explain the rationale behind the different agent numbers used in various experiments. Please also provide a more detailed analysis of the success rate decline, and, if possible, include quantitative measures of the decline rate or additional data points to better illustrate the trend.

---

> ### Author Response · Authors · 2024-11-24
> **Rebuttal**
>
> We thank the reviewer for their thoughtful feedback and constructive comments. We appreciate the recognition of the novelty of our approach, its contributions to addressing the critical challenges of distributed decision-making with privacy and scalability guarantees, and the strengths of our experimental evaluations. Below, we provide detailed responses to the weaknesses and questions raised.
>
> ---
>
> ## Response to Weaknesses
> ### Explanation of Tables 1, 2, and 3
> We appreciate the reviewer’s feedback regarding the clarity of the explanations for Tables 1, 2, and 3. To clarify:
> - In Table 1, 2, and 3, the **first row** shows the number of agents $N$ for each set of experiments ($N = 10, 20, 40, 80$).
> - The **second row** illustrates the communication topology among agents.
> - The **third row** contains **three small tables**, each corresponding to a separate experimental run for the given topology. Each small table has rows equal to the number of agents $N$, representing the energy states of all agents in the network.
> - From these results, we observe that:
>   1. In each experimental run, **all agents in the network converge to the same equilibrium**, confirming alignment of their strategies.
>   2. Across different experimental runs, the **converged equilibria differ**, highlighting the framework’s ability to achieve diverse equilibria based on the initialization and randomness in the alignment process.
> ---
>
> ## Response to Questions
>
> ### 1. What Happens if There Are No Neighbors Within Communication Range?
> - When an agent has no neighbors within communication range, it cannot compute the gradient of its energy state, leading to a stall in strategy alignment for that agent. This is analogous to the scenario discussed in **Section 5.4: Robustness Against Lost Connections**, where high communication loss rates result in agents forming isolated groups, hindering their ability to achieve a local minimum collectively.
> However, in practice:
>   - Agents are typically deployed with known communication constraints, ensuring adequate connectivity for coordination tasks.
>   - For edge cases, fallback mechanisms such as temporary centralized coordination or relay agents can be used to maintain system performance.
>
> ### 2. Shared Random Number Generator and Privacy Implications
> - While all agents share the same random number generator (RNG), the seeds used for generating disruptive noise are **local to each agent pair**. Specifically, let $\mathcal{G}$ and $\mathcal{F}$ denote the RNG and seed generation function shared by all agents, respectively.
>   - For the $p$-th alignment process, each agent $i$ generates a seed $sed_{ij}^p=\mathcal{F}(\mathbf{W}_{ij}, p)$ for the communication with the agent $j$.
>   - During the $t$-th energy minimization step of the $p$-th alignment process, the agent $i$ computes the disruptive noise $\sigma_{ij}=\mathcal{G}(sed_{ij}^p, t)$ used for the communication with the agent $j$.
>   - Note that only the agent $i$ and agent $j$ know $\mathbf{W}_ {ij}$. Adversaries cannot reconstruct the noise sequence $\{\sigma_{ij}\}$ unless this information is leaked. For more discussion, please see the analysis regarding an opponent with $L3$ ability in Section 4.3.
>
> ### 3. Robustness of Noise Addition Against Persistent Adversaries
> - The disruptive noise $\sigma_{ij}$ used for obfuscating communication data is computed as $\sigma_{ij} = \mathcal{G}(sed_{ij}^p, t)$, where $sed_{ij}^p$ and $t$ vary across alignment processes and energy minimization steps, respectively.
> - The probability of the noise $\sigma_{ij}$ being the same for different $(p, t)$ is extremely low due to the differences in $sed_{ij}^p$ and $t$. Moreover, assuming two coordination processes indexed by $p_1$ and $p_2$ achieve converged states in $t_1$ and $t_2$ steps, the probability that both processes have the same converged states and identical noise $\sigma_{ij}$ is negligible.
> - Therefore, it is almost impossible that the same converged states lead to the same communication data. This ensures that even persistent adversaries with knowledge of all previous communication data cannot infer whether the current converged energy state has appeared before.
>
> ### 4. Clarification of Time Units in Table 4
> We thank the reviewer for pointing out the missing time unit declaration. The time units in Table 4 are nanoseconds (ns). In real-world applications, the duration of each operation impacts the speed and resource consumption of the strategy alignment process. Compared to AES, our method demonstrates computational efficiency, consuming an order of magnitude less time. This efficiency contributes to the method’s environmental friendliness and practicality. We will update Table 4 to include the time unit.

---

> ### Author Response · Authors · 2024-11-24
> **Rebuttal-cont.**
>
> ### 5. Rationale Behind Different Agent Numbers and Success Rate Decline
> - **Reason for Different Agent Numbers**:
>   The number of agents in different experiments was chosen based on time constraints and specific analysis goals:
>   - For **Tables 1–3**, we conducted 12 experiments per table (4 settings of $N$ and 3 alignment processes). For **Figure 3**, we ran 200 experiments (4 settings of $N$, 5 seeds, 10 loss rates), requiring over five times the computational resources of Tables 1–3.
>   - For Figure 3, we selected $N=30$ instead of $N=80$ to maintain a linear progression in the number of agents and to better evaluate its impact on the relationship between link loss rate and alignment success rate.
> Nevertheless, we are running experiments with $N=80$ for Figure 3 and will update the results later.
>
> - **Analysis of Success Rate Decline**:
>   - From Figure 3, the success rate decline becomes more apparent as the number of agents increases, mainly due to communication loss and increased difficulty in maintaining alignment across a larger network.
>   - For example, at a $0.7$ loss rate, the success rate decreases from over $0.9$ ($N = 10$) to under $0.5$ ($N = 40$), illustrating that larger networks experience sharper declines in success rate. This is expected because, at the same loss rate, larger networks are more likely to contain isolated agents, amplifying the decline.
>   - Therefore, for $N = 80$, we expect an even sharper decline, with significant drops occurring at lower loss rates (e.g., $<0.6$).
>
> ---
>
> We appreciate the reviewer’s thoughtful comments and believe these clarifications and updates will address the concerns raised. Thank you again for your constructive feedback and recognition of the contributions and strengths of our work.

---

> > ### Comment · Reviewer_Sc7y · 2024-11-27
> >
> > I thank you for the explanations, but I am still not fully convinced by your approach:
> > 1. when there are no agents in the communication range you said it is the same as when communication is lost. Unfortunately. when communication is lost the agents cannot a local minimum, so the method proposed fails and no way to recover is proposed in the paper.
> > 2. the seeds for the random number generation are shared by pairs of agents, so if an agent is in communication with multiple agents it needs to keep track of multiple seeds. I do not recall reading about this in the paper.
> > 3. the robustness to noise addition against adversaries is not guaranteed. You said that it is almost impossible to end up with the same communication data, but it is still possible, and this reduces the guarantees of the proposed approach.
> > 4. the experimental results presented have the agents equally distributed along the circumference of the area, which is a simple topological structure, but there are no results with agents placed in random locations, which might be more likely to get out of communication range.
> >
> > I will keep my original score.

---

> ### Author Response · Authors · 2024-11-29
> **Rebuttal for new questions**
>
> We thank the reviewer for their additional feedback and for raising these important points. Below, we provide detailed responses to address the concerns:
>
> ### **1. When There Are No Agents Within Communication Range**
> - We appreciate the reviewer’s concern about scenarios where agents lose communication. Like many distributed systems, our framework relies on communication between agents. If an agent becomes fully isolated—losing all communication with others—it is indeed theoretically infeasible for that agent to align with others, as it cannot receive the necessary information.
> - However, if the agent retains partial communication, strategy alignment may still be achievable with the assistance of relay agents that forward information for it. While this specific recovery mechanism is outside the scope of this paper, we acknowledge its importance and recognize it as a valuable direction for future research and its potential relation with the problem of network failure restoration.
>
> ### **2. Tracking Multiple Seeds for Random Number Generation**
> - Thank you for raising this question about seed management. We would like to clarify that the seed $sed_{ij}^p$ used for communication between agents $i$ and $j$ during the $p$-th alignment process is generated from the weight $\mathbf{W} {ij}$ which is shared between the two agents, i.e., $sed_{ij}^p = \mathcal{F}(\mathbf{W} {ij}, p)$. Therefore, agent $i$ does not need to track multiple seeds. Instead, it only needs to store $\\{\mathbf{W} {ij} \mid j=1,\dots,N,$ and $A_{ij}=1\\}$, which is already required for computing the gradient of its energy states. This design avoids additional overhead while ensuring efficient operation.
>
> ### **3. Robustness of Noise Addition Against Adversaries**
> - We appreciate the reviewer’s observations about the robustness of noise addition. The communication data after adding noise is represented as $\mathbf{W} {ij}\rho(\mathbf{u}_i^t) \oplus \mathcal{G}(sed _{ij}^p, t)$. Even if two communication data instances are identical, there are two possible scenarios:
>
>    - Both the energy states and the noise are identical: $\mathbf{W} {ij}\rho(\mathbf{u}_i^{t1}) = \mathbf{W} {ij}\rho(\mathbf{u}_i^{t2})$ and $\mathcal{G} (sed _{ij}^p, t1) = \mathcal{G}(sed _{ij}^p, t2)$.
>
>    - The energy states and noise differ, but their bitwise XOR operations coincidentally result in the same output: $\mathbf{W} {ij}\rho(\mathbf{u}_i^{t1}) \neq \mathbf{W} {ij}\rho(\mathbf{u}_i^{t2})$ and $\mathcal{G} (sed _{ij}^p, t1) \neq \mathcal{G}(sed _{ij}^p, t2)$.
>
> - Therefore, given the same communication data, the adversary cannot determine whether the converged states and noise are the same or if the identical communication data is simply the result of coincidental XOR operations. We believe this ambiguity protects the privacy robustness of our approach.
>
> ### **4. Experimental Results with Random Agent Placement**
> - Thank you for your observation regarding the presentation of agent placement in our experiments. The agents are displayed as equally distributed along a circle primarily for illustrative purposes, as this arrangement simplifies the visualization and makes the communication topology more easily interpretable. Plotting agents with random locations can result in a visually cluttered graph, which may obscure the relationships between nodes and their connections.
> - However, we assure the reviewer that our experimental evaluations are not limited to this specific topology. We have conducted experiments using three types of network topologies—random, small-world, and ring structures. The small-world topology, where agents have only sparse connections, approximates scenarios where agents might fall out of the communication range of others. These experiments are included in our evaluation and demonstrate the robustness of our approach across various configurations.
>
> ---
>
> We hope these responses address the concerns raised and provide greater clarity for our work. Thank you again for your valuable feedback and for helping us refine and improve our paper.

---

> > ### Comment · Reviewer_Sc7y · 2024-11-30
> >
> > Thanks for the additional clarifications, which help me understand what you proposed. Unfortunately, I have read what was in the paper. For instance, I did not see any mention that the experimental results shown with agents on a circle are not all the experiments you did.  The notation used is not easy to follow and the writing is dense. Even Fig 1, meant to illustrate the processes, is hard to understand because the notation used is not clearly defined.
> > I cannot increase my score because the writing needs significant improvements, otherwise readers will be confused like I have been. I do not object to accept the paper, but the outcome depends on the opinions of the other reviewers.

---

> > > ### Author Response · Authors · 2024-11-30
> > > **Response to Reviewer’s Feedback**
> > >
> > > We sincerely thank the reviewer for their additional clarifications and for taking the time to review our responses. We appreciate the opportunity to address the points raised and provide further clarity.
> > >
> > > - **Clarification of Experimental Evaluations**
> > >    - We regret that our previous response "our experimental evaluations are not limited to this specific topology" may have caused some misunderstandings regarding the scope of our experimental evaluations. To clarify, regarding the effectiveness experiments, Table 1, 2, and 3 (in the original version) or Table 1, 2, and 5 (in the revised version) represent all the experiments we conducted. These experiments correspond to three different topologies: random, small-world, and ring.
> > >    - The agents are plotted as equally distributed along a circle purely for illustrative purposes, as this arrangement makes the communication topology more visually interpretable. These topologies allow agents to be positioned randomly, as long as the communication links (i.e., edges of the topologies) remain valid. We hope this explanation resolves any confusion and provides a clearer understanding of the experiments presented.
> > >
> > > - **Acknowledgment of Dense Notation and Writing**
> > >    - We understand and acknowledge the reviewer’s concern about the dense writing and extensive use of notation. Since this work introduces the new Federated Coordination problem and analyzes various properties of our proposed method, some complexity in presentation is unavoidable.
> > >    - That said, we have made efforts to address this issue in the revised version by including a detailed notation table in Appendix A.1 to assist readers. Additionally, we plan to refine the text and provide clearer explanations in the final version of the paper, should it be accepted, to further improve readability and accessibility.
> > >
> > > - **Regarding the Score**
> > > We respect and accept the reviewer’s decision to maintain the current score. We are already very grateful for your kind support and valuable feedback, which have significantly helped us refine and improve our work : ) Your constructive comments and suggestions will undoubtedly contribute to making this paper clearer and more impactful.
> > >
> > > ---
> > >
> > > Thank you again for your thoughtful feedback and for engaging deeply with our work. We appreciate your time and effort in reviewing our paper.

---

### Official Review · Reviewer_bEza · 2024-11-04

**Soundness:** 3
**Presentation:** 3
**Contribution:** 3
**Rating:** 6
**Confidence:** 4

**Summary:**

This paper proposed a new problem - federated coordination, where decentralized agents such as drones choose their strategies to minimize energy values. Each agent only needs to gather information from their immediate neighbors, which contributes to the privacy protection and efficiency of the coordination. Besides, the scenario where cooperative agents face an opponent is modeled by a one-shot adversarial team game, which is a zero-sum game. Specifically, strategies are stored first before the coordination. In the proposed algorithm, agents will update their energy values based on the neighbor's information until reaching the minimum by gradient descent. Furthermore, they developed an encryption-free yet privacy-preserving communication protocol, to further maintain the confidentiality of agents' strategies. Last, experiments were conducted to show the validation and efficiency of the methods and algorithms.

**Strengths:**

1. The problem setting is novel, which addresses both privacy and coordination without a centralized agent.
2. The algorithm is computationally efficient as it converges at sub-linear speed.
3. The example of drones is good for illustration purposes.
4. The figures for the experiment results are great for visualizing the outcomes of this work.

**Weaknesses:**

As mentioned in this work, "with random W and b, determining all local minima of the corresponding energy-based model is very difficult". Therefore, the work uses a preset collection of local minima and then determines the value of W and b. However, these local minima may not be readily accessible in practice, which could limit the approach's effectiveness.

**Questions:**

In reality such as the drones, how could you decide or find the preset collection of local minima?

---

> ### Author Response · Authors · 2024-11-24
> **Rebuttal**
>
> We thank the reviewer for their detailed feedback and thoughtful comments. We appreciate the recognition of the novelty of the problem setting, the computational efficiency of our algorithm, and the clarity of our experimental results. Below, we address the identified weaknesses and questions to further clarify and strengthen our contributions.
>
> ---
>
> ## Response to Weaknesses
>
> ### 1. Accessibility of Local Minima
> - We understand the concern regarding the accessibility of local minima in practice. However, the primary goal of our framework is not to ensure that every local minimum in the energy landscape is readily accessible but rather to guarantee that **each strategy in the given set of coordination strategies** has a non-zero probability of being aligned during coordination rounds. This ensures the diversity of potential coordination strategies.
> - To achieve this, we use **Algorithm 2**, which ensures that the number of local minima $L$ is larger than the size of the coordination strategy set $S$. By associating each strategy with at least one local minimum, we ensure that every strategy in the set of coordination strategies has a non-zero probability of being aligned during the coordination process.
> - However, due to the complexity of the energy landscape, the probability of each strategy being aligned may vary, and some strategies could have a relatively small probability of alignment. To address this potential concern, we can easily adjust **Algorithm 2** to enforce a larger ratio between $L$ and $S$. Specifically, by modifying Line 5 in Algorithm 2 from "**while** $L(M^*, \xi) < S$ **do**" to "**while** $L(M^*, \xi) < qS$ **do**", where $q$ is the desired ratio, we can ensure that each strategy corresponds to approximately $q$ local minima. This adjustment increases the likelihood of each strategy being associated with a local minimum whose probability of being aligned is relatively large.
> - These practical considerations ensure that all strategies in $S$ remain accessible, preserving the diversity of potential coordination strategies. We will include this clarification and the optional adjustment to Algorithm 2 in the final version of the paper to address this concern comprehensively.
>
> ---
>
> ## Response to Questions
>
> ### 1. How to Determine the Preset Collection of Local Minima in Practice?
>
> In practice, the preset collection of local minima is determined through systematic algorithmic processes. Specifically, we follow these steps:
>
> 1. **Using Algorithm 2**:
>    - Algorithm 2 computes the values of $\xi$, $M$, and $L=2^{M-\xi}$ based on the adjacency matrix of the agents and the size of the given set of strategies $S$. It ensures that the number of local minima $L$ is greater than $S$. This guarantees that each strategy can be associated with at least one local minimum, enabling all strategies to be accessible during coordination strategy alignment.
>    - If a higher likelihood of strategy alignment is required, we can modify Algorithm 2 to enforce a larger ratio between $L$ and $S$ (e.g., $L \geq qS$ where $q > 1$), ensuring each strategy corresponds to multiple local minima.
>
> 2. **Generating the Local Minima**:
>    - Once $L$, the required number of local minima, is determined using Algorithm 2, **binary coding** is employed to generate the corresponding preset collection of local minima. This approach systematically creates energy states that can serve as local minima, ensuring they are well-distributed across the energy landscape and sufficient in number to guarantee that all strategies are readily accessible.
>
> These steps ensure that the practical implementation aligns with the theoretical requirements of the framework and guarantees diversity in coordination strategies across rounds.
>
> ---
>
> We appreciate the reviewer’s insightful comments and believe that our rebuttal will address the concerns raised. Thank you again for recognizing the strengths and contributions of our work.

---

> > ### Comment · Reviewer_bEza · 2024-11-26
> >
> > I want to thank the authors for their responses to clarify my questions in detail. I will update the scores.

---

> > > ### Author Response · Authors · 2024-11-27
> > >
> > > We sincerely appreciate the reviewer’s thoughtful consideration of our responses and the decision to update the score. Thank you for your constructive feedback, and we look forward to addressing any additional concerns during further discussions.

---

### Official Review · Reviewer_5pGE · 2024-11-04

**Soundness:** 3
**Presentation:** 3
**Contribution:** 2
**Rating:** 5
**Confidence:** 3

**Summary:**

Authors present an approach to ensure multi-agent decentralized coordination using only local (neighbors) information and preventing information leakage, based on defining a joint strategy energy equation of which any desired coordination strategy is a local minimum. An algorithm for designing appropriate local minima is presented, and the effectiveness is demonstrated in coordination experiments.

**Strengths:**

- The solution of achieving decentralized coordination through a common potential function and carefully designed local minima is interesting, and relatively novel to the best of my knowledge.
- The paper is very clear and ideas are well presented.

**Weaknesses:**

- My main concern is that the paper may fall slightly on the edge of suitable topics for ICLR. While multi-agent learning is clearly relevant, in this case the work feels more suitable for a multi-agent or robotics venue, where much of the current literature on multi-agent coordination is presented.
- Although I believe the method is novel, there are many existing works on decentralized coordination for multi-agent systems, as well as decentralized consensus control for distributed systems. Additionally I do not see such a clear distinction between decentralized coordination and federated coordination as the authors seem to argue for. A more thorough positioning of the paper with respect to existing literature would help. This affects the relevance and contribution.

**Questions:**

- Would be interesting to see how this method applies to dynamic games or more general multi agent planning problems, both theoretically and experimentally.

I do not have any other major questions. I believe the paper is good and the ideas are interesting. My assessment scores are entirely based on a combination of contribution weight and appropriateness of the venue.

---

> ### Author Response · Authors · 2024-11-24
> **Rebuttal**
>
> We sincerely thank the reviewer for their time and thoughtful feedback. We appreciate the acknowledgment of the novelty and clarity of our work. Below, we address the main points of concern and provide additional context to strengthen the case for the paper's relevance and contribution.
>
> ---
>
> ## Response to Weaknesses
>
> ### 1. Relevance to ICLR
> While we acknowledge that this work could also fit within multi-agent or robotics venues, we believe it is well-suited for ICLR due to its methodological innovations and its contributions to learning energy model parameters, which serve as representations of the given set of coordination strategies. Moreover, prior ICLR publications have also explored areas related to multi-agent coordination, further underscoring the relevance of this work to the conference. Examples include:
> - **"Emergent Communication through Negotiation"** (Cao, K., Lazaridou, A., et al., ICLR 2017): The authors explore how agents can develop communication protocols through negotiation, leading to coordinated behaviors without pre-defined languages.
> - **"Emergent Coordination Through Competition"** (Liu, S., Lever, G., Merel, J., et al., ICLR 2019): The authors study emergent coordination mechanisms driven by competitive multi-agent interactions.
> - **"Discovering Generalizable Multi-agent Coordination Skills from Multi-task Offline Data"** (Zhang, F., Jia, C., et al., ICLR 2023): The authors study how to improve generalization in cooperative multi-agent reinforcement learning (MARL) across multiple tasks, especially when agents have to adapt to new tasks with varying agents and targets using only offline data from limited tasks.
>
>
> ### 2. Positioning Relative to Existing Literature
> We appreciate the reviewer’s suggestion to strengthen the positioning of our work. Our proposed **federated coordination** framework addresses a **unique problem** that distinguishes it from existing approaches in decentralized coordination and consensus. The key differences are as follows:
>
> 1. **Decentralized Coordination**:
>    - Existing decentralized coordination works primarily focus on **learning a coordination strategy** through frameworks like multi-agent reinforcement learning (MARL). However, the learned strategy is typically **stationary**, making it vulnerable in adversarial settings. An adversary can train a best-response strategy against the stationary learned strategy, significantly reducing its effectiveness.
>    - In contrast, federated coordination ensures that agents align their strategies from a **set of predefined strategies** in a **decentralized and random manner**, making it unpredictable for adversaries. This dynamic alignment mechanism inherently prevents adversaries from effectively countering the strategy.
>
> 2. **Decentralized Consensus**:
>    - Traditional decentralized consensus ensures that agents agree on the **same value**, which can correspond to a coordination strategy. However, this approach has a critical vulnerability: a **single compromised agent** can leak the agreed-upon strategy to an adversary, enabling the adversary to select the best response.
>    - Federated coordination overcomes this issue by preserving the **privacy of the aligned strategy**, even in adversarial settings. The privacy-preserving protocol ensures that the adversary cannot determine which strategy is being used, thus preventing it from taking a best response.
>
> These distinctions highlight that federated coordination is not merely an extension of decentralized coordination or consensus but rather a novel framework designed specifically for adversarial multi-agent environments. It introduces privacy as a critical dimension, alongside decentralized strategy alignment, to address challenges that are not effectively handled by existing methods.
>
> In the revised paper, we will expand the related work section to emphasize these distinctions and better position federated coordination as a significant and novel contribution to the multi-agent and machine learning literature.
>
>
> ---
>
> ## Response to Questions
>
> ### 1. Application to Dynamic Games and General Multi-Agent Planning
> - We agree that extending our framework to dynamic games and multi-agent planning problems is an exciting direction for future work. Our current focus is on laying the foundation for federated coordination in static settings, which we believe is a necessary step before tackling more complex and dynamic environments as federated coordination is a brand-new problem setting.
> - Theoretically, our energy-based model could be extended to dynamic games by incorporating temporal dependencies in the energy function, potentially using a time-evolving graph structure. Experimentally, this would require modifications to handle dynamic strategy updates while maintaining privacy. We will add these ideas to the discussion as future work.
>
> ---
>
> Thank you again for recognizing the novelty and clarity. We hope our rebuttal clarifies our work's relevance and contribution.

---

> ### Comment · Reviewer_5pGE · 2024-11-26
> **Rebuttal acknowledgement**
>
> I thank the authors for the clear reply and addressing my concerns. I will change my score to reflect this. I still have some concerns over the appropriateness of the venue / contributions but these will be subject to discussion with other reviewers, and I'm happy to revise the assessment further.

---

> ### Author Response · Authors · 2024-11-26
> **Response to rebuttal acknowledgement**
>
> We sincerely thank the reviewer for taking the time to carefully consider our responses and for updating the score. We appreciate the constructive feedback and are grateful for the opportunity to address any remaining concerns during further discussions.

---

### Official Review · Reviewer_Kthc · 2024-11-07

**Soundness:** 3
**Presentation:** 3
**Contribution:** 3
**Rating:** 6
**Confidence:** 3

**Summary:**

This paper proposes a new problem formulation called "federated coordination" which is geared towards decentralized strategy alignment with privacy preservation in multi-agent systems. Strategy alignment among agents is posed as an optimization problem using energy based models (EBMs), where an undirected graph is used to represent the interaction among neighboring or communicating agents. Different strategies correspond to different local minima for the EBM and the communication protocol among the agents is designed to be privacy-preserving against eavesdropping and information leakage without depending on more computationally demanding cryptography techniques. In an adversarial game setting between strategic agents, this paper proposes an algorithm for distributed strategy alignment against an adversary and demonstrates the scalability of the algorithm with different team sizes, different graph connectivity and robustness to communication loss.

**Strengths:**

1. The problem setting proposed in this paper is novel and interesting with possible applications to multi-agent systems in robotics, security games, etc.

2. The paper is generally well-written.
- The authors have pointed out the differences in their proposed problem setting compared to prior work in multi-agent reinforcement learning, game theory and differential privacy.
- The two parts of the proposed algorithm: strategy alignment and privacy preservation against adversarial opponents, is clearly explained with discussion of the implications of the design choices under different settings (eg. in sec 4.3).

3.  Framing strategy alignment as equivalent to finding the local minima in an EBM for a particular inter-agent interaction layout is a principled and flexible approach. The iterative communication based gradient computation to minimize the agent's energy function naturally leads to a decentralized alignment procedure without a central coordinating agent.

**Weaknesses:**

While I appreciate the experiments outlined in support of the proposed problem formulation and the algorithm, the authors do not sufficiently highlight the potential weaknesses of the proposed approach and the scope for improvement.
1. The proposed algorithm requires a pre-determined mapping between an agent's energy at a possible local minima to the corresponding action as part of the joint strategy. This reduces the difficulty of the setup by only focusing on achieving the local minimum starting from a random initialization of the agent's energy. For the algorithm to be actually practical, perhaps the mapping should be dynamic and learned from inter-agent interaction.
2. In this paper, the authors assume a pre-defined set of strategies and alignment means each agent looking up from the dictionary exactly which strategy they are meant to follow, which is different from the multi-agent reinforcement learning setting where the focus is primarily on learning a joint strategy for the team. By jointly optimizing both the energy landscape and learning high-reward strategies for the team, it might improve robustness against strategic adversary and unreliable communication.
3.  Instead of binary values for the local minima in the energy landscape, allowing continuous values would make the experimental setup more practical and flexible. The experiments also do not follow an adversarial team game setting which was used to motivate the federated coordination framework.

**Questions:**

1. Can you explain what is meant in line 527-528, especially what does inability to reach alignment mean in terms of outcome from an adversarial team game?

2.  How was $\rho(\cdot)$ selected?

3. Please add a notation table to clearly define all the variables used in the paper.

---

> ### Author Response · Authors · 2024-11-24
> **Rebuttal**
>
> We thank the reviewer for their detailed and constructive feedback. We appreciate the recognition of the novelty and importance of the proposed problem formulation and the strengths of our work, including the principled approach using energy-based models (EBMs), the clarity of our algorithm, and the well-motivated problem setting. Below, we address the specific weaknesses and questions raised in the review:
>
> ## Response to Weaknesses
> ### 1. Pre-determined Mapping Between Energy and Actions
>
> As the first work for solving this challenging problem, we focus on establishing a baseline framework and validating its effectiveness through extensive experiments. While our current implementation uses a pre-determined mapping between energy states and actions to simplify the setup for clarity and proof of concept, we agree that future research can explore dynamic and learned mappings for greater flexibility and adaptability.
>
> ### 2. Comparison to Multi-Agent Reinforcement Learning (MARL)
>
> The distinction between our approach and MARL is intentional: federated coordination focuses on decentralized alignment for pre-defined strategies without centralized training, whereas MARL emphasizes learning joint strategies. Nevertheless, integrating strategy learning into the framework, as suggested, is an intriguing idea. We will expand our discussion to include this as a future direction, emphasizing the complementary nature of MARL and our work.
>
> ### 3. Binary vs. Continuous local minima
>
> We use binary values for the local minima as this simplifies the determination of the number of local minima, $L$ (Algorithm 2). Theoretically, the energy model supports the use of continuous values for the local minima. However, implementing this in practice would require refining Algorithm 2 to accommodate the additional complexity.
>
> ### 4. Adversarial Team Game Experiments
>
> We conducted experiments to demonstrate the effectiveness of the proposed framework, specifically its ability to achieve distributed strategy alignment. The adversarial team game setting is primarily related to the privacy of the aligned strategy and we address this concern in Section 4.3, Analysis of Privacy-Preserving Capability.
>
> ## Response to Questions
> ### 1. Line 527-528: Inability to Reach Alignment
>
> The inability to reach alignment in our context means that agents fail to converge to a shared local minimum in the energy landscape because the break of communication links makes some agents be isolated, i.e., they cannot exchange information necessary for the state update. In an adversarial team game, this implies that the cooperative agents might not execute a coordinated strategy, reducing their utility against adversaries.
>
> ### 2. Selection of $\rho(\cdot)$
>
> The activation function $\rho(\cdot)$ was chosen as the hyperbolic tangent function (tanh) for its desirable properties in the context of the proposed energy-based model. Specifically, the tanh function saturates at −1 and +1, ensuring that the energy values remain bounded. This boundedness is crucial for stabilizing the energy minimization process and for guaranteeing the existence of local minima in the energy landscape.
>
> ### 3. Notation Table
>
> We appreciate the suggestion to add a notation table. This is a great idea to improve clarity. We provide the table below and will incorporate it in the revised version to define all variables systematically.
>
> ---
>
> Thank you again for your thoughtful and detailed feedback. We believe our rebuttal and revisions will further strengthen the manuscript and address the concerns raised.

---

> ### Author Response · Authors · 2024-11-24
> **Rebuttal-notation table**
>
> ## Notation Table
>
> | Symbol          | Definition                                                                                 |
> |------------------|-------------------------------------------------------------------------------------------|
> | $ N $         | Number of cooperative agents in the system.                                               |
> | $ \mathcal{A} $         | Joint action space of all agents.                                                         |
> | $ \mathcal{B} $         | Set of actions available to the adversary.                                                |
> | $ U(\mathbf{a}, b) $   | Utility function, where $ \mathbf{a} \in \mathcal{A} $ and $ b \in \mathcal{B} $.                                   |
> | $ E(\mathbf{u}) $      | Energy function used to represent the system of agents.                                    |
> | $ \mathbf{u}_i $       | Energy state of agent $ i $.                                                            |
> | $ \mathbf{v}^l $       | Local minimum of the energy function.                                                     |
> | $ \rho(x) $   | Activation function (tanh in this work).                                                  |
> | $ \mathbf{W}_{ij} $    | Weight matrix for the edge connecting agent $ i $ and agent $ j $.                    |
> | $ b_i $       | Energy state bias of agent $ i $.                                                       |
> | $ \epsilon $  | Convergence threshold for energy minimization.                                            |
> | $ \delta_{ij} $ | Kronecker delta, equal to 1 if $ i = j $, otherwise 0.                                 |
> | $ \mathcal{G}(s, t) $   | Random noise generator function with seed $ s $ and step $ t $.                       |
> | $ \mathcal{F}(W, p) $   | Seed generation function based on weight $ W $ and alignment process $ p $.           |
> | $ S $         | Set of predefined coordination strategies.                                                |
> | $ L $         | Number of local minima in the energy landscape.                                           |
> | $ M $         | Dimensionality of the energy state vector $ u_i $.                                      |
> | $ \otimes $   | Outer product operator.                                                                   |
> | $ \oplus $    | Bitwise XOR operation.                                                                    |
> | $ \nabla E(\mathbf{u}) $ | Gradient of the energy function with respect to the energy states $ u $.              |

---

> > ### Comment · Reviewer_Kthc · 2024-11-27
> >
> > Thank you for the response. I would recommend including in the main paper this discussion from the rebuttal of the current limitations and potential avenues for future work, especially since this work is being proposed as a baseline to define a new setup. I would like to maintain my current score.

---

> > > ### Author Response · Authors · 2024-11-29
> > >
> > > Thank you for your thoughtful feedback and recommendation. We have uploaded a revised version of the paper that includes a discussion of the current limitations and potential avenues for future work in the main paper, as suggested. We appreciate your constructive input and consideration.

---

### Official Review · Reviewer_GvyX · 2024-11-07

**Soundness:** 3
**Presentation:** 3
**Contribution:** 2
**Rating:** 5
**Confidence:** 5

**Summary:**

This paper addresses a new problem named “federated coordination”, where the goal is for distributed agents to align their strategies, while maintaining their privacy and decentralized computations/communication. The proposed methodology for tackling this problem relies on an energy-based optimization problem formulation. In particular, the core idea is to associate predefined strategies with predetermined local minima in this problem and arrive to random strategies alignments based on the random local minima this optimization will converge to. Privacy requirements are met through a simple communication protocol that can address three types of threats. The presented experiments results suggest that the proposed methodology is effective for the problem that is studied.

**Strengths:**

The strengths of this submission are listed as follows:

1. An interesting problem/methodology of associating predefined multi-agent strategies with prescribed local minima in an energy-based model minimization is presented.

2. A distributed algorithm for solving this energy-based optimization problem is provided.

3. Privacy requirements are considered by showing how the proposed algorithm can handle three different types of privacy threats.

4. The experiments suggest that the proposed algorithm is indeed effective for addressing the problem introduced in this paper. In particular, its performance, scalability, computational efficiency and robustness to lost connections are validated.

**Weaknesses:**

The weaknesses of this submission are outlined as follows:

1. Although this is an interesting problem to study, the authors should better justify its meaningfulness and importance. For example, a motivating application for multi-drone systems is provided in the beginning of Section 3, stating that “To optimize their performance, the drones must randomly switch between these strategies for each operation, ensuring unpredictability in their tactics”. The authors should provide references where arriving to completely random and environment/opponent independent - yet predefined - tactics is indeed useful.

2. The claim in lines 191-192, i.e., “Given $\mathbf{W}$ and $\mathbf{b}$, the size of local minimum collection $L$ and the value of each $\mathbf{v}^l$ are uniquely determined” requires further clarification/justification. The problem presented in Eq. (1) is highly non-convex and as (correctly) explained in the first paragraph of Section 4.1, “determining all local minima of the corresponding energy-based model is very difficult, if not impossible”. As a result, these two parts of the paper appear to contradict each other. The authors should either better explain/rephrase their claim in lines 191-192 or provide a theoretical justification.

3. The authors claim in Section 4.1 that their algorithm “ensures that all local minima are known”. To my best understanding, solving Eq. (4) will indeed give a pair ($\mathbf{W},\mathbf{b}$) that sets the prescribed local minima $\mathbf{v}^l$ as local minima of the optimization in Eq. (1). Nevertheless, how does this guarantee that there are no other local minima in this problem in addition to the prescribed ones?

4. For the proposed approach to make sense, the authors assume that the system in Eq. (4) has a unique solution (see claim “This ensures that $\mathbf{W}$ and $\mathbf{b}$ are uniquely determined by the constraints”). Nevertheless, is there a guarantee that this system is solvable? If the constraints are more than the d.o.f., can’t it be the case that there is no solution for this system? My expectation is that this also has to do with the linear independence of the equations in system (4).

5. It is not clear why the authors opt to solve the least-squares minimization in Eq. (6) instead of directly solving the system in Eq. (4). Note that if the optimal value of the problem in Eq. (6) is other than $J^* = 0$, then (4) is not satisfied, and as a result the prescribed local minima $\mathbf{v}^l$ are not going to be actual local minima of the problem in Eq. (1). Now, whether the optimal cost can be 0 or not is directly related to whether system (4) is solvable – see previous point. As a result, the motivation for employing the minimization presented in Eq. (6) is unclear.

6. At the end of Section 4.1, an algorithm for setting $L$ and $M$ is motivated and then fully described in the Appendix. The motivation for this algorithm appears to be that “setting $L$ and $M$ can result in an unsolvable Eq. (6)”. However, it is unclear how this algorithm indeed guarantees the solvability of the system. To my best understanding, the algorithm presented in appendix A.3 computes the optimal *user-defined number of local minima* $L$ and state size $M$, under the constraints in lines 781-783 and the cost that $M$ and $L$ should be as small as possible. Nevertheless, this approach does not take into account important aspects such as the fact that i) the number of local minima is actually related to the form of $\mathbf{W}, \mathbf{b}$ - which are going to emerge later after solving Eq. (4) and ii) the linear independency of the resulting equations in Eq. (4) is a major factor of whether the system is solvable.

7. Given the importance of arriving to a (prescribed) local minimum using the distributed algorithm presented in Section 4.2, it is crucial to provide a guarantee that the proposed algorithm indeed converges to such a point. If this is not the case, then converging to a point other than the prescribed minima could be dangerous – especially for applications such as the coordination of multi-drone teams which is the motivation for this paper. Despite its expected importance, such a theoretical guarantee is not provided in this submission.

8. This paper's motivation is to achieve randomness on selecting between the predefined strategies. Nevertheless, whether this is in fact achieved is not clearly explored either theoretically or empirically.

    a. From a theory perspective, it would be interesting if the authors can establish a guarantee on this “randomness”. My expectation is that this would depend on the initializations, the underlying local optimizers and of course $\mathbf{W}, \mathbf{b}$. For example, assuming the initializations are coming from uniform distributions, can any conclusion be drawn on perhaps some bounds on the probabilities of each strategy to emerge?

    b. If such a theoretical result is too mathematically demanding to prove, then the authors should at least provide an empirical validation of their claim. An easy way to verify this could be testing whether "scanning" the space of initial conditions for $\mathbf{u}$ provides solutions that do not disproportionately favor the selection of specific strategies over others, or that do not make specific strategies to never appear.

**Questions:**

The authors are encouraged to address the following questions:

1. Pointing to weakness (1), can the authors better justify why this is a meaningful problem to address by providing specific references, for example on the motivation application, i.e., multi-drone systems, where their problem setup would be directly applicable?

2. Pointing to weakness (2), can the authors provide a clarification on the point made in lines 191-192 and especially what they mean with all local minima being “uniquely determined”?

3. Can the authors address the concern raised in weakness (3)? Is there a guarantee that the pair $(\mathbf{W}, \mathbf{b})$ we get from solving Eq. (4) will not lead to an optimization problem in Eq. (1) that has extra local minima in addition to the prescribed ones $\mathbf{v}^l$?

4. Pointing to weakness (4), why is it guaranteed that the system in Eq. (4) is solvable? How can it be avoided that this system might have no solution? Shouldn’t this also be related to the linear independence of these equations?

5. Pointing to weakness (5), can the authors elaborate on opting to solve the minimization in Eq. (6) instead of the system in Eq. (4)? If the optimal value of $J$ in Eq. (6) is other than 0 wouldn’t this mean that  system (4) is not satisfied and as a result the prescribed local minima are not actual local minima?

6. Can the authors address the comments raised in weakness (6), regarding whether the algorithm in appendix A.3 actually guarantees the solvability of the system?

7. Pointing to weakness (7), can it be proven that the algorithm proposed in Section 4.2 is guaranteed to converge to one of the prescribed local minima?

8. Pointing to weakness (8), would it be possible that the authors elaborate on whether this desired “randomness” of strategy selection is indeed achieved. If providing a theoretical result that guarantees that is too mathematically tedious, would it be possible that it is at least shown empirically in the experiments section?

9. In the beginning of Section 5, it is mentioned that the Adam optimizer is used for solving the optimization in Eq. (6) as well as the local optimizations of the agents using the gradients in Eq. (8).

    a. Regarding using Adam to optimize Eq. (6), why is there a need to employ stochastic optimization to solve this problem? How is stochasticity incorporated here? Does it refer to a notion of evaluating at random subsamples (minibatches) of the “data” provided in Eq. (5)? The authors should elaborate on that and why Adam is preferred over standard or accelerated (non-stochastic) gradient descent, especially since this is also a convex minimization.

     b. It would be useful if the authors could also provide details on how Eq. (8) is used with Adam as the local optimizer of each node and whether again stochasticity is incorporated through evaluating at random subsamples.

10. In Section 5.2, to my best understanding, the authors refer with number of steps $Z$ to the rounds of the distributed optimization in Algorithm 1? Nevertheless, the amount of performed local Adam iterations is not stated. Isn’t the number of local Adam iterations affecting the number of rounds/steps $Z$ needed to achieve convergence?

11. Similarly, in the computational efficiency analysis in Section 5.3, how many local Adam iterations are performed per step?

---

> ### Author Response · Authors · 2024-11-29
> **Rebuttal**
>
> We sincerely thank the reviewer for their detailed and constructive feedback, as well as for recognizing the strengths of our work, including the introduction of a novel problem, the development of a distributed algorithm for energy-based optimization, and the validation of its effectiveness through experiments. We also apologize for the delay in responding to your thorough review, as we are currently managing a busy schedule. In our revised submission, we have discussed several key issues raised in the review, including the presence of spurious local minima and the solvability of Eq. (4), in the end of Section 4.1. To further clarify these points, we have added discussions in Appendix A.6 and A.7, where we detail how our implementation addresses these concerns. We hope these revisions effectively address your concerns and improve the clarity and rigor of our work. Below, we provide detailed responses to address your concerns.
>
> ### **1. Response to weakness 1/question 1**: Justifying the importance of random and predefined tactics in multi-drone systems.
>
> We thank the reviewer for pointing out the need to better justify the meaningfulness and importance of the problem. Below, we provide additional references and context to support the use of random switching strategies in multi-agent systems.
>
> - **Coordinating Randomized Policies for Increasing Security of Agent Systems** [1]: This study highlights the problem where "an adversary can learn or observe the patrolling strategy and use this to its advantage." It demonstrates the effectiveness of randomized patrolling strategies for ensuring unpredictability in systems like the ARMOR system at LAX, preventing adversaries from exploiting deterministic patterns.
> - **A Robust Hybrid Iterative Learning Formation Strategy for Multi-UAV Systems with Multi-Operating Modes** [2]: This study emphasizes the advantages of mode switching in multi-UAV systems, particularly in enhancing robustness and flexibility under disturbances or communication failures.
>
> These studies support the application of random switching strategies to improve performance and robustness in multi-agent systems. However, both works propose centralized methods, which depend on centralized infrastructures and pose privacy risks at the centralized point. Our decentralized approach addresses these challenges, providing privacy-preserving and distributed strategy alignment.
>
> [1] Paruchuri, Praveen, et al. "Coordinating randomized policies for increasing security of agent systems." Information Technology and Management 10 (2009): 67-79.
>
> [2] Yang, Song, et al. "A Robust Hybrid Iterative Learning Formation Strategy for Multi-Unmanned Aerial Vehicle Systems with Multi-Operating Modes." Drones 8.8 (2024): 406.
>
> ### **2. Response to weakness 2/question 2**: Clarifying the claim regarding the unique determination of local minima.
>
> We thank the reviewer for raising this important point. The claim in lines 191–192 asserts that $\mathbf{W}$ and $\mathbf{b}$ are the parameters defining the energy landscape $E(u)$ (Eq. 1), which consequently determine the existence of local minima. However, determining the specific number of local minima $L$ and the value of each local minimum $\mathbf{v}^l$ requires additional computation.
>
> The statement in Section 4.1, “determining all local minima of the corresponding energy-based model is very difficult, if not impossible,” highlights the computational challenges associated with the high dimensionality of the state space—an issue commonly referred to as the curse of dimensionality. For our model, determining all local minima would require searching in a continuous space of dimension $M^N$. These two points are not contradictory but rather address different aspects of the problem.
>
> To address this challenge, our approach constructs $\mathbf{W}$ and $\mathbf{b}$ such that the resulting energy landscape includes pre-determined local minima $\\{\mathbf{v}^l\\}$. This enables the alignment of the energy-based model with desired outcomes without necessitating the exhaustive identification of all local minima.

---

> ### Author Response · Authors · 2024-11-29
> **Rebuttal-cont1**
>
> ### **3. Response to weakness 3/question 3**: Guaranteeing that prescribed local minima are the only local minima.
>
> We acknowledge that while our solution ensures the prescribed local minima correspond to those of the constructed energy function, it does not theoretically eliminate the possibility of spurious minima—points that are local minima of the constructed energy function but do not belong to the set of prescribed local minima. To address this, our implementation employs binary coding, as described in Section 5.1 and Appendix. Below, we detail the concept behind Algorithm 2 and explain how strategy alignment benefits from binary coding.
>
> 1. **Idea of Algorithm 2**:
>    - Using binary coding to represent preset local minima, the number of preset minima $L$ is given by $2^M$, where $M$ is the dimension of the energy states. The goal of Algorithm 2 is to find the minimum $M$ that satisfies the following conditions:
>      1. The number of constraints $LNM$ equals or exceeds the degrees of freedom in $\mathbf{W}$ and $\mathbf{b}$, i.e., $\frac{M^2}{2} \sum_{i,j} A_{ij} + NM$ (Lines 4 and 7 in Algorithm 2).
>      2. $L \geq S$, where $S$ is the number of strategies (Line 5 in Algorithm 2).
>    - A straightforward approach would increment $M$ until both conditions are met. However, as $M$ increases, the constraints $NM \cdot 2^M$ grow super-exponentially, while the degrees of freedom increase polynomially. This may cause $LNM$ to become excessively large relative to $\frac{M^2}{2} \sum_{i,j} A_{ij} + NM$.
>    - To mitigate this, Algorithm 2 introduces a variable $\xi$ to control the growth of $LNM$. Specifically, given the state dimension $M$, $\xi$ entries of the state are fixed across the minima, reducing $L$ to $2^{M-\xi}$. Algorithm 2 iteratively finds the minimum $M$ (Lines 4 and 7) and a suitable $\xi$ (Lines 5–8) to ensure: $2^{M-\xi}NM \geq \frac{M^2}{2} \sum_{i,j} A_{ij} + NM$ and $2^{M-\xi} \geq S$.
>
> 2. **Alignment Based on Binary Coding**:
>    - Once Algorithm 2 determines the appropriate $M$ and $\xi$, we construct $2^{M-\xi}$ local minima by:
>      - Setting the first $M-\xi$ entries of each state to either $3$ or $-3$.
>      - Fixing the remaining $\xi$ entries at $3$.
>    - The prescribed local minima are then used to compute $\mathbf{W}$ and $\mathbf{b}$ based on Eq. 6. During strategy alignment, we determine which preset minimum the agents converge to by evaluating whether each entry of their energy state is closer to $3$ or $-3$. This ensures that **any spurious minima are reliably mapped to the prescribed ones**, maintaining alignment integrity.
>
> 3. **Empirical Validation**:
> - We conducted extensive empirical tests to verify the effectiveness of this implementation. These include detailed experiments across various topologies, as shown in Tables 1–3 in Section 5.1, and trials that evaluate the scalability. In all cases, the converged states consistently aligned with the predetermined minima, demonstrating the effectiveness of our implementation.
>
> We appreciate the reviewer’s insightful question and hope this explanation clarifies how our implementation addresses the potential presence of spurious minima.

---

> ### Author Response · Authors · 2024-11-29
> **Rebuttal-cont2**
>
> ### **4. Response to weakness 4/question 4**: Solvability of the system in Eq. (4).
>
> - The existence of solutions to Eq. (4) is mathematically intricate due to implicit constraints imposed by the topology $A$ and the predetermined set of local minima $\{\mathbf{v}^l\}$. Specifically, the block matrix $\overline{\mathbf{W}}$ defined in Eq. (5) is subject to implicit constraints determined by the topology $A$. To elucidate these constraints, consider a scenario where the inter-agent weights are homogeneous, i.e., $\mathbf{W} {ij} = \mathbf{W}_{i'j'}$ for all indices $i, j, i', j'$. In this case, the block matrix can be represented as a Kronecker product, $\overline{\mathbf{W}} = A \otimes \mathbf{W}$, implying that $\det(\overline{\mathbf{W}}) = \det(A) \det(\mathbf{W})$. If the communication network is singular, indicated by $\det(A) = 0$, then the block matrix is inherently singular, resulting in $\det(\overline{\mathbf{W}}) = 0$. Consequently, Eq. (4) becomes unsolvable.
>
> - In more general cases as outlined in Eq. (5), the relationship involving $\overline{\mathbf{W}}$ becomes more complex. There is no straightforward method to address solvability in the most general form, thus we employ an optimization-based approach, utilizing $J(\mathbf{W},\mathbf{b})$ to assess the problem's solvability.
>
> - In this approach, $\overline{\mathbf{W}}$ is treated as a parameter within a linear system. The solution to Equation (4) can fall into one of three categories: (1) infinitely many solutions, (2) a unique solution, or (3) no solution, contingent on the relationship between the degrees of freedom (d.o.f) and the number of constraints. In both cases (1) and (2), the optimal value of $J^*(\mathbf{W},\mathbf{b})$ is 0, while in case (3), $J^*(\mathbf{W},\mathbf{b})$ is greater than 0.
>
>      - Case (1): To eliminate this scenario, Algorithm 2 is employed to ensure that the number of constraints is at least equal to the d.o.f. Subsequently, a small Gaussian noise (standard deviation$<0.0001$) is added to each preset binary-coded local minimum. This guarantees the linear independence of the equations specified by Eq. (3), thereby eliminating case (1).
>
>     - Case (2): This is the desired outcome. Note that adding Gaussian noises to the preset local minima results in $\mathbf{W}$ and $\mathbf{b}$ that cause the real local minima to deviate slightly from the preset values, e.g., a preset minimum of 3 may result in real minima of 3.00005, 3.0001, etc. However, since the converged state is mapped to binary encoding, this ensures alignment with the preset local minima.
>
>    - Case (3): In this instance, an optimal solution, denoted as $\mathbf{W}^*$ and $\mathbf{b}^*$, can still be identified by minimizing the least-squares error $J(\mathbf{W}, \mathbf{b})$. This solution does not exactly satisfy Eq. (4), and $J(\mathbf{W}^*, \mathbf{b}^*)$ quantifies the deviation of $\{\mathbf{v}^l\}$ from the true local minima determined by $\mathbf{W}^*$ and $\mathbf{b}^*$. Given the convexity of $J$, the loss is distributed among $\{\mathbf{v}^l\}$. A small $J(\mathbf{W}^*, \mathbf{b}^*)$ (average loss per local minimum < 0.1) indicates proximity of true local minima to each preset $\mathbf{v}^l$. The binary coding mapping ensures that the true minimum aligns with the preset minimum. Conversely, a large $J(\mathbf{W}^*, \mathbf{b}^*)$ suggests that the current communication topology is inadequate for achieving alignment, prompting adjustments to the topology prior to system deployment.
>
> ### **5. Response to weakness 5/question 5**: Justifying the use of least-squares optimization in Eq. (6).
>
> We chose least-squares minimization due to its computational efficiency and robustness to numerical noise, which is particularly useful in real-world applications. Moreover, as noted in our response to Weakness 4, deriving a general solution to Eq. (4) is complex. Optimizing the value of $J(\mathbf{W}, \mathbf{b})$ in Eq. (6) can provide a practical way to determine the solvability of Eq. (4).

---

> ### Author Response · Authors · 2024-11-29
> **Rebuttal-cont3**
>
> ### **6. Response to weakness 6/question 6**: Algorithm in Appendix A.3 guaranteeing solvability.
>
> We thank the reviewer for their thoughtful observation regarding the solvability of the system under the proposed algorithm for setting $L$ and $M$. Below, we address the concerns raised and provide further clarifications.
>
> - As mentioned in our response to Weakness 3, **Algorithm 2** in Appendix A.3 determines the appropriate $M$ and $\xi$ to satisfy the constraints $2^{M-\xi}NM \geq \frac{M^2}{2} \sum_{i,j} A_{ij} + NM$ and $2^{M-\xi} \geq S$. These conditions ensure that the degrees of freedom in $\mathbf{W}$ and $\mathbf{b}$ are adequately constrained, while the number of prescribed minima remains sufficient to support strategy alignment.
>
> - To address the potential issue of linear independence in Eq. (4), we introduce **small perturbations** to the preset binary-coded local minima through Gaussian noise. This step ensures that the constraints remain linearly independent, preventing issues of degeneracy during optimization.
>
> - These procedures collectively ensure that the resulting $\mathbf{W}$ and $\mathbf{b}$ from solving Eq. (6) construct an energy landscape capable of supporting effective strategy alignment. Specifically:
>    - The energy landscape either contains local minima with a very small deviation from the prescribed minima (case (2) in our response to Weakness 4) or minima that are close to the prescribed minima (case (3) in our response to Weakness 4).
>    - In both cases, binary coding reliably maps converged energy states to the prescribed minima, guaranteeing alignment of strategies.
>
> Finally, extensive empirical testing has validated this approach, with no observed exceptions. In all trials, every converged state corresponded to a prescribed local minimum, demonstrating the effectiveness of the proposed method.
>
> We hope this clarifies how our framework ensures solvability and robustness in constructing the energy landscape. Thank you again for the opportunity to address this important point.
>
>
> ### **7. Response to weakness 7/question 7**: Guaranteeing convergence to prescribed local minima.
>
> As mentioned in our response to Weakness 3, we employ binary-coded preset local minima to ensure that any converged state is reliably mapped to the prescribed minima. While we acknowledge that a full theoretical guarantee is not currently provided, extensive empirical testing supports the reliability of our approach. Across hundreds of trials, all converged states consistently aligned with the prescribed minima, demonstrating the robustness and effectiveness of our implementation.
>
> ### **8. Response to weakness 8/question 8**: randomness in strategy selection.
>
> We thank the reviewer for their insightful comments regarding the randomness in selecting between predefined strategies.
> The randomness in our approach arises from the initialization of states, which are sampled from a uniform distribution, and the selection of the optimizer. While we acknowledge that a full theoretical analysis of the probabilities for each strategy to emerge is mathematically demanding, we are addressing this concern empirically. Specifically, we set $N=20$, with the state dimension $M=15$ and the number of invariant dimensions $\xi=5$ computed by Algorithm 2. This resulted in $L=2^{M-\xi}=1024$ binary-coded local minima. We ran the alignment process 35,596 times and recorded the number of convergences for each local minimum. The results are as follows:
>
> - **6 local minima** had converged times between 20–25.
> - **108 local minima** had converged times between 25–30.
> - **398 local minima** had converged times between 30–35.
> - **362 local minima** had converged times between 35–40.
> - **129 local minima** had converged times between 40–45.
> - **21 local minima** had converged times between 45–50.
>
> #### Key Observations and Implications
> 1. **All Local Minima Accessed**: Importantly, all 1024 binary-coded local minima were accessed at least six times. This confirms that every predefined strategy had a non-zero probability of being selected.
> 2. **Reasonable Randomness**: While some variation in convergence frequencies exists, this is expected due to stochastic initialization. The convergence frequencies indicate that the majority of preset local minima were reached 30~40 times and only a few minima had converged times more than 45 or less than 25. The results suggest that the method does not disproportionately favor a small subset of local minima, ensuring randomness in strategy selection.
>
> These findings empirically support the claim that our approach achieves randomness in selecting predefined strategies while maintaining coverage of all local minima. We hope this addresses the reviewer’s concerns and further demonstrates the robustness and effectiveness of the proposed framework. Thank you again for highlighting this important aspect, which has strengthened the evaluation of our work.

---

> ### Author Response · Authors · 2024-11-29
> **Rebuttal-cont4**
>
> ### **9. Response to question 9 on Adam Optimization**:
>
> We thank the reviewer for their thoughtful questions regarding the use of the Adam optimizer in solving Eq. (6) and Eq. (8). Below, we provide detailed clarifications.
>
> - **Use of Adam for Solving Eq. (6)**:
>    - Adam is employed in our work for its superior convergence speed, particularly in large systems. While Adam is often referred to as a stochastic optimizer, its stochasticity arises when a function is optimized using data that is randomly sampled across multiple steps (e.g., minibatches in machine learning). However, this is not applicable in the case of solving Eq. (6), where:
>      - All constraint data (i.e., the preset local minima) is provided at the beginning of the optimization process.
>      - This setup ensures that the optimization is deterministic due to the given data and the convex nature of the problem.
>    - Adam is preferred over standard or accelerated gradient descent for its adaptive learning rate mechanism, which often results in faster convergence without requiring manual tuning of hyperparameters.
>
> - **Use of Adam for Solving Eq. (8)**:
>    - In contrast to Eq. (6), the optimization problem in Eq. (8) is non-convex. This makes Adam particularly advantageous for accelerating convergence, especially near saddle points, where standard gradient descent methods may slow down or fail to escape effectively.
>
> We hope this clarifies the rationale behind our choice of the Adam optimizer and its application to Eqs. (6) and (8). We appreciate the opportunity to elaborate on these details.
>
> ### **10. Response to question 10**:
>
> We thank the reviewer for raising this question regarding the relationship between the number of rounds $Z$ in Algorithm 1 and the local Adam iterations.
>
> It seems there may have been some misunderstanding regarding the usage of the Adam optimizer during the strategy alignment process in Algorithm 1. In each step of the strategy alignment, every agent follows the procedures specified in lines 7–10 of Algorithm 1. Specifically:
> - In line 9, each agent $i$ computes the gradient of $\mathbf{u}_i^t$ based on Eq. (8).
> - In line 10, the agent inputs the computed gradient $\frac{\partial E(\mathbf{u}^t)}{\partial \mathbf{u}_i^t}$ and its current state $\mathbf{u}_i^t$ into the Adam optimizer to update $\mathbf{u}_i^t$ as $\mathbf{u}_i^{t+1}$.
>
> Thus, in every step of the strategy alignment, there is exactly one Adam operation performed by each agent. Consequently, when the total number of distributed optimization rounds is $Z$, the number of local Adam iterations for each agent is also $Z$.
>
> We hope this clarifies the usage of the Adam optimizer in our strategy alignment process.
>
>
> ### **11. Response to question 11**:
>
> As noted in our response to Question 10, there is exactly one local Adam operation performed per step for each agent. The computational efficiency analysis in Section 5.3, however, focuses on the encryption and decryption times for communication between neighboring agents. The time spent updating internal states using Adam is not included in this analysis, as the primary objective is to compare the communication overhead between AES and EFC, rather than the computational cost associated with internal state updates.
>
> We hope this clarifies the scope of the computational efficiency analysis.
>
> ---
>
> We deeply value the time and effort the reviewer has invested in providing these detailed and constructive comments, which have significantly improved our work.

---

> > ### Comment · Reviewer_GvyX · 2024-12-03
> >
> > Thank you very much for your detailed responses. I appreciate that the authors have addressed all of my concerns and made the relevant additions in the revised paper.
> >
> > Consequently, I am raising my score from 3 to 5 (weak reject). The reasons I am still not recommending this paper for acceptance are the following:
> >
> > 1. The problem setup addressed by this work is very specific. It is hard to argue about the significance and broader potential impact of this work if there are not sufficient applications to benefit from it.
> > 2. The paper is on the border of being suitable for a ML conference. I am unsure what portion of the ML community would find this work directly relevant.

---

> > > ### Author Response · Authors · 2024-12-03
> > >
> > > We sincerely thank you for your detailed feedback and for raising your score. We greatly appreciate the time and effort you have taken to review our work thoroughly and provide constructive comments, which have significantly helped us improve our paper.
> > >
> > > ### **Regarding the Problem's Significance and Broader Impact**
> > >
> > > We understand your concern about the specificity of the problem setup and the difficulty in arguing for its broader impact. While Federated Coordination may appear specific, we believe it addresses a growing need in decentralized multi-agent systems where privacy, scalability, and adversarial robustness are critical. Examples include:
> > >
> > > 1. **Multi-Drone Systems**: As cited in the revised version, dynamic and unpredictable coordination strategies are crucial for robust performance in adversarial scenarios, such as surveillance, disaster response, and security operations. These real-world applications highlight the practical value of our work.
> > >
> > > 2. **Decentralized Networks Beyond Drones**: The principles of Federated Coordination extend to other domains, such as decentralized resource allocation, distributed sensor networks, and swarm robotics. These systems often operate in adversarial environments where opponents may attempt to disrupt coordination or exploit deterministic patterns in strategies. Our work addresses this challenge by enabling agents to align on strategies in a decentralized manner, ensuring robustness against such adversarial threats.
> > >
> > > By introducing this framework, we aim to lay the foundation for further research that could generalize Federated Coordination to other settings and applications.
> > >
> > > ### **Relevance to the ML Community**
> > >
> > > We acknowledge the concern about the paper's suitability for an ML conference. While our work introduces a new problem formulation and solution in Federated Coordination, it is deeply rooted in the ML principles of:
> > >
> > > 1. **Energy-Based Models (EBMs)**: The core of our approach leverages EBMs, a class of probabilistic models extensively studied in the ML community, to define and solve the coordination problem.
> > >
> > > 2. **Optimization and Distributed Learning**: The proposed method incorporates centralized optimization to determine $\mathbf{W}$ and $\mathbf{b}$, and distributed optimization to align local minima, areas that are highly relevant to ML researchers focused on multi-agent systems and decentralized learning.
> > >
> > > 3. **Future ML Research Directions**: Our work opens exciting avenues for future ML research, such as:
> > >    - **Distributed Learning in Dynamic Environments**: When applying Federated Coordination to dynamic and complex environments, it may require the distributed learning of topology or strategies to counter dynamic adversaries. This has great potential to attract the attention of the ML community.
> > >    - **Learning Dynamic Mapping Between Strategies and Local Minima**: Developing models to dynamically learn and optimize mappings between strategies and local minima could improve flexibility and robustness, aligning with ongoing research interests in the ML community.
> > >
> > > By presenting our work at an ML venue, we aim to engage the community in addressing emerging challenges at the intersection of ML, multi-agent systems, and decentralized coordination.
> > >
> > > ### **Final Remarks**
> > >
> > > We are grateful for your recognition of our efforts in addressing your concerns and incorporating relevant revisions into the paper. We hope the above clarifications address the concerns regarding the problem’s significance and the paper’s relevance to the ML community. Thank you again for your thoughtful feedback and for raising your score—it means a lot to us.

---

### Meta-Review · Area_Chair_bRBR · 2024-12-23

**Metareview:**

This paper studies a new problem called “federated coordination”, where the goal is to achieve decentralized strategy alignment across distributed agents, while maintaining the privacy of strategy choices. The proposed methodology relies on an energy-based optimization problem formulation. The idea is to associate predefined strategies with predetermined local minima, and privacy requirements are met through a simple communication protocol that can address three types of threats.  Experiment results are also presented.

The main weakness of this paper, as raised by the reviewers, is its relevance to the ML community. While multi-agent learning is relevant to ICLR, it is unclear whether the audiences of ICLR would find this submission interesting, as it might fall slightly on the edge of suitable topics for ICLR. This work might be more suitable for a multi-agent or robotics venue, where much of the current literature on multi-agent coordination is presented.

Moreover, as mentioned by the reviewers, the proposed algorithm requires a pre-determined mapping between energy and actions of the agents. This might limit the practicality of the proposed approach.

Given the high standards of ICLR and the weakness mentioned above, I would recommend rejecting this paper.

**Additional Comments On Reviewer Discussion:**

The reviewers raised concerns regarding the problem's significance and broader impact, its relevance to the ML community, as well as the requirement of a pre-determined mapping between energy and actions of the agents. Although the authors provided responses which addressed some of those concerns, concerns regarding the relevance of this work to the ML community remain.

---

### Decision · Program_Chairs · 2025-01-22

Reject